# Structure-Activity-Relationship and Mechanistic Insights for Anti-HIV Natural Products

**DOI:** 10.3390/molecules25092070

**Published:** 2020-04-29

**Authors:** Ramandeep Kaur, Pooja Sharma, Girish K. Gupta, Fidele Ntie-Kang, Dinesh Kumar

**Affiliations:** 1Sri Sai College of Pharmacy, Manawala, Amritsar 143001, India; poojasharmagndu@gmail.com (R.K.); pooja.sharma2007@yahoo.co.in (P.S.); 2Department of Pharmaceutical Sciences and Drug Research, Punjabi University, Patiala 147002, India; 3Department of Pharmaceutical Chemistry, Sri Sai College of Pharmacy, Badhani, Pathankot 145001, India; girish_pharmacist92@rediffmail.com; 4Department of Chemistry, Faculty of Science, University of Buea, P.O. Box 63 Buea, Cameroon; 5Institute for Pharmacy, Martin-Luther-Universität Halle-Wittenberg, Kurt-Mothes-Str. 3, 06120 Halle (Saale), Germany; 6Institut für Botanik, Technische Universität Dresden, Zellescher Weg 20b, 01062 Dresden, Germany

**Keywords:** AIDS, anti-HIV, natural products, MOAs

## Abstract

Acquired Immunodeficiency Syndrome (AIDS), which chiefly originatesfroma retrovirus named Human Immunodeficiency Virus (HIV), has impacted about 70 million people worldwide. Even though several advances have been made in the field of antiretroviral combination therapy, HIV is still responsible for a considerable number of deaths in Africa. The current antiretroviral therapies have achieved success in providing instant HIV suppression but with countless undesirable adverse effects. Presently, the biodiversity of the plant kingdom is being explored by several researchers for the discovery of potent anti-HIV drugs with different mechanisms of action. The primary challenge is to afford a treatment that is free from any sort of risk of drug resistance and serious side effects. Hence, there is a strong demand to evaluate drugs derived from plants as well as their derivatives. Several plants, such as *Andrographis paniculata*, *Dioscorea bulbifera*, *Aegle marmelos*, *Wistaria floribunda*, *Lindera chunii*, *Xanthoceras sorbifolia* and others have displayed significant anti-HIV activity. Here, weattempt to summarize the main results, which focus on the structures of most potent plant-based natural products having anti-HIV activity along with their mechanisms of action and IC_50_ values, structure-activity-relationships and important key findings.

## 1. Introduction

Acquired immunodeficiency syndrome (AIDS) is a disease of the cell-mediated immune system or T-lymphocytes of the human body. In AIDS, the count of helper T cells isreduced, which directly stimulates the production of antibodies from B-cells. Consequently, the body’s natural defense system against AIDS infection isdestroyed [1]. According tothe World Health Organization (WHO), it is estimated that about 75 million individuals have been infected from the human immunodeficiency virus (HIV), and about 37 million people are still under the fighting stage. The prevalence of HIV is expected to increase significantly due to illiteracy, non-hygienic living conditions, unsafe sexual relationships and lack of awareness [2]. Initially, the first human retrovirus was founded at the National Cancer Institute, in the USA by Robert Galloand his colleagues, after being first discovered in 1981 among homosexuals. In 1983, Professor Luc Montagnier and co-workers later discovered the AIDS virus at the Institute Pasteur, in Paris [3]. In 1986, the International Committee on Viral Nomenclature were the first to officially name the AIDS virus the human immunodeficiency virus (HIV) [4]. Currently, Africans worldwide are stricken by this illness more than any other race [5].

### 1.1. The HIV Structure

The motor agent for AIDS is an animal retrovirus named HIV that is ready to replicate and integrate its infectious DNA into the host cell’s healthy DNA. It is an animal virus that chiefly attacks the body’s helper T cells [6,7]. The virus is spherically shaped, having a diameter of around 90–120 nm. Its genetic material generally contains a single standard RNA fiber metameric into two similar fibers and is connected with an enzyme called reverse transcriptase (RT). The viral coating contains a lipid bilayer that is derived from the membrane of the host cells and spikes of glycoprotein that are like projecting knob. It consists of two protein coats, as depicted in Figure 1 [8,9]. Internally, the virus contains a protein layer (the matrix), which consists of the necessary proteins and nuclear material. The virus also contains an enzyme known as a protease that disintegrates the viral polyproteins to form new functional proteins. The role of reverse transcriptase is to catalyse the conversion of the viral RNA into viral DNA and integrase, which allows the entry of viral DNA into the host nucleus [10,11].

### 1.2. Replication Cycle of HIV

The complete HIV replication cycle is represented in Figure 2. After the entrance of the virus into the body of an individual, the virus invades the body cells through CCRS or CXCR4 receptors shown on the top of the macrophages, known as T-lymphocytes, dendritic cells and monocytes [6,7]. After entering the host cell, the virus binds with chemokine receptors and interacts with cell membrane proteins. The virus then releases and utilizes its reverse transcriptase (RT) enzyme for the synthesis of viral DNA from its viral genome i.e., HIV RNA. This conversion allows the virus to enter into the host cell nucleus, where the enzyme integrase releases and perform integration of its viral DNA into the host cell’s DNA [8,9,10,11]. Newly formed HIV proteins and viral RNA shifts towards the cell membrane and reuniteswith immature HIV. The new immature (non-infectious) virus then buds off from the host cells, which inturn initiates the release of the protease enzyme from the viruses that cause the breakdown of long-chain polypeptides of immature viruses. The newly formed small protein particles make the new mature viruses that enter into the new host cells for spreading the infection [12,13].

### 1.3. Diagnosis

The level of HIV infection is diagnosed from the blood plasma of the host through their viral RNA mass estimation. The infection has been associated with the period of acute symptoms viz; lymphadenopathy, fever, weight loss, lethargy, general malaise, pharyngitis, rashes, nausea, headache, myalgias and meningitis, etc. [2,6]. During acute HIV infection, the viral RNA is at the highest levels in the blood plasma. It is estimated that the amount and characteristics of the virus indicate its pathogenesis and replication. Hence, the clinical details and infection progression depend on the host characteristics, along with the viral genotype [14,15,16]. ELISA and Western Blotting were the two main tests employed for the diagnosis of AIDS in the past. ELISA is used for the detection and measurement of the antibodies that are produced against a specific pathogen [17], while Western Blotting was employed for confirmation of ELISA positive tests. It is used to check the specific proteins in the blood sample. The samples go through the protein denaturation and then gel electrophoresis. The combined effect of both tests is found to be 99% accurate. Nowadays, various advantageous alternatives are available in place of Western Blotting.Among the advantages associated with such alternatives is less time-consuming testing [18]. 

### 1.4. Present Therapy for HIV/AIDS

The knowledge of HIV has beenmade public since the 1980s. However, there is currently no availability of efficacious therapy or vaccine for the entire destruction of the virus [2]. Current AIDS treatments have many drawbacks, e.g., complex and tedious treatment protocols, requiring expertise from medical practitioners, solid motivation and patient’s commitment. Antiretroviral therapy (ART) is only about twenty years old, meaning that further approaches are still in progress. The usage of certain medications can slow the progress of the disease without the patient necessarily being promised total recovery. However, with the development of new entities and immune modulators, it is now feasible to fight this deadly disease [19,20,21]. The drugs provide a meaningful advancement in mitigation, control, cure, and prevention. With the establishment of highly active antiretroviral therapy (HAART) and anti-retroviral agents in 1996 decreased the mortality and morbidity of AIDS has been observed. Antiretroviral therapy is presently prescribed for all adult patients living with HIV [2]. Many types of combination approaches such as the use of nucleoside reverse transcriptase inhibitors, fusion inhibitors, non-nucleoside reverse transcriptase inhibitors, integrase inhibitors, protease inhibitors, together with immunomodulators have been prescribed to achieve a proficient therapeutic response [3,6,9,10,11]. Due to the lack of non-accessible effective regimens, it has been noted that the objective of therapy is to sturdily and maximally prohibit viral replication so that the individual can achieve and maintain an adequate immune response against the potential viral pathogens. The higher the abolition of viral replication, the lower the incidence of development of the drug-resistant virus. The minimization in the mortality and morbidity of the disorder has turned it from a lethal syndrome to a chronic and controllable situation [5,19,21]. It is now advised that all HIV positive individuals with the perceptible virus, disregarding their count of CD4 cells, should be recommended with ART quickly after diagnosis, to avoid further progression [20].

### 1.5. Drawbacks of Current Anti-Retroviral Therapy

Even though it is impressive to deal with all the symptomatic and asymptomatic HIV infected persons, no long-lasting clinical outcomes have been illustrated in asymptomatic patients with acceptable immune competency [22]. Arguments in contrast to early remedies in asymptomatic patients involve the dangerous side effects of anti-HIV drugs, their toxicity and destructive effect of anti-HIV drugs on quality of life, the possibility of drug resistance restricting future treatment opportunities, big cost, drug interactions, the limited capability of available regimens, failure of treatment [23,24,25]. The right time to start anti-HIV therapy remainsuncertain. The antiviral drugs that act on the HIV also affect the host cells; they may harm the host cell’s nuclear material along with the HIV genome. With nucleoside reverse transcriptase inhibitors, toxicity is primarily due to the partial provision of cellular DNA polymerase. Neutropenia and Anaemia are extremely critical and dose-dependent adverse effects. Moreover, to date, there is no vaccine or cure for HIV infection, and the efficacy of antiretroviral therapy consist of a combination of two or three antiviral agents, targeting different steps of the virus replication cycle, can be compromised by the selection of strains resistant to one or multiple drug classes and current treatment-associated toxicity. However, these drugs have only limited or transient clinical benefit due to their noxious side effects and the emergence of viral variants resistant to HIV-1 inhibitors. Unfortunately, their use is limited due to the speedy emergence of resistant viral strains and to the severe toxic side effects. Hence, new natural products can be considered as novel leads for the development of new effective and selective anti-AIDS agents [24,25,26].

## 2. Plants with Anti-HIV Potential

Presently, strategies available to combat AIDS are restricted by the evolution of multidrug resistance. That is why novel targets and new efficacious drugs are required for achieving the goal of an entire eradication of AIDS. Also, infected cells persist and constitutea basic barrier to the elimination of HIV-1. For the past 10 years, the mechanism by which the virus persists hasnecessitated anovel pathway in the discovery of new drug compounds that work efficaciously against HIV without activating the T cells of the immune system [27,28]. To attain this goal, it has been recommended by the WHO that ethnomedicines and various other natural constituents should be systematically tested to combat HIV [29,30]. Interestingly, in the 1990s, natural products with their mechanisms against HIV-1 enzymes like reverse transcriptase, integrase, protease and some fusion inhibitors were discovered. The natural drugs have chemical diversity with higher hit rates in high throughput screening and high capability to reach the target site [31,32,33]. Several alkaloids, flavonoids, coumarins, terpenoids, and polyphenolic compounds, aswell as known therapeutic agents having an array of biological activities like anticancer, analgesics, anti-inflammatory and exert anti-HIV activity extracted from various plants, were found [34,35,36,37,38,39,40,41,42,43,44]. These became the sources of inspiration for many research activities, e.g., the anti-HIV potential of components of *Dioscorea bulbifera* [45], *Euphorbia sikkimensis* [46], *Culendula officinalis* [47], *Sceletium tortuosum* [48], *Brazilian propolis*, *Kadsura lancilimba*, *Lithocarpus litseifolius*, and *Ocimum labiatum* [49,50,51,52].

Taken together, the present review highlights the discovery of plant-based molecules during the last few decades that have been used in the management of HIV. A detailed account of plants according to their mechanism of action and activity of secondary metabolites has been discussed. In addition to the structures of most potent phytochemicals, mechanistic insights revealed during the biological evaluation, IC_50_ values and important key findings have also been presented. The detailed mechanisms of this action and structure-activity-relationships of some of the compound classes remain to be further investigated. This assemblage will be of great help for the scientific community working towards the development of anti-HIV drugs. In this review, the natural medicinal plants are described in two categories:Plants according to their mechanism of action.Plants according to the activity of secondary metabolites.

### 2.1. Natural Plants According to Their Mechanism of Action

Therapeutic agents of natural origin may be an encouraging alternative solution for the treatment of several disorders and conditions [53,54,55,56,57,58,59]. In anti-HIV research, attention is chiefly paid tocompounds which interfere with several steps involved in the HIV replication process. For example, almost all the anti-HIV drugs act against the viral proteins represented by the viral protease, integrase, and reverse transcriptase [60]. Anti-HIV drugs can be classified into several groups according to their action on the life cycle of HIV [61]. Hence, different drugs act on these different steps of replication and inhibit the further expansion of the virus into the body. A group of researchers reported the activities of HIV-PR inhibitors from different plants primarily divided into the following categories [62,63,64,65,66,67,68,69,70,71]:(a)Fusion inhibitors (FI)(b)Reverse transcriptase inhibitors (RTI)(c)Integrase inhibitors (ITI)(d)Protease inhibitors (PRI)(e)Immunomodulators(f)Antioxidants

#### 2.1.1. Fusion Inhibitors

Fusion inhibitors are also known as Entry inhibitors. These are mainly CCR5 co-receptor antagonists which inhibit the binding of HIV surface glycoproteins with the host cell’s receptor [72]. Infection primarily starts with the binding of the viral gp120 to the CD4 cell receptor expressed on the surface of T cells, macrophages, and some monocytes. This results in the conformational change which further stimulates the interaction of secondary gp120 with co-receptor CCR5 [73]. FIs prevent the entry of the virus into the host cell by inhibiting the fusion of virus particles with the membrane of the host cell, which is the early first step of virus replication [74].

Phytoconstituents from some plants, like *Listeria ovate, Cymbidium hybrid, Hippeastrum hybrid, Epipactis helleborine* and *Urtica dioica* possessing the activities of fusion inhibitors and act against the HIV-1 and HIV-2 [75,76]. Matsuda et al. reported an alkaloid Cepharanthine (**1**) isolated from *Stephania cepharantha* having anti-HIV and anti-tumour potential without exerting any type of serious toxic effects. This compound modifies the plasma membrane fluidity and prevents viral cell fusion [77]. A diterpene lactone named Andrographolide (**2**) shown in Figure 3 was obtained from the herb *Andrographis paniculata* and possesses HIV-1 fusion inhibition propertiesevaluated in vitro using AZT (Zidovudine) as a positive control [78,79,80,81,82]. Several other derivatives have been derived synthetically to exert more potent anti-HIV properties [83,84].

#### 2.1.2. Plant Extracts as Reverse Transcriptase Inhibitors

The HIV virus utilizes the reverse transcriptase enzyme for the conversion of its viral RNA into DNA. RT inhibitors mainly act upon this enzyme and prohibit one of the essential steps of viral replication [85,86]. Several natural products have been isolated from plants are available in theliterature, which have been screened for their activity against RT [66]. The plants which tested positively for reverse transcriptase inhibition include; *Culendula officinalis*, *Acacia mellifera*, *Uvaria angolensis*, *Hypericum scruglii*, *Spaganium stoloniferum*, *Calophyllum brasiliense*, *Maytenus buchanani*, *Prunus Africana*, *Vernonia jugalis*, *Maytenus senegalensis*, *Melia azedarach*, *Calophyllum inophyllum*, *Lomatium suksdorfii*, *Coriandrum sativum*, *Chrysanthemum morifolium* and *Swertia franchetiana* [47,66,67,68,69,70,71,72,73,74,75,76,77,78,79,80,81,82,83,84,85,86,87,88,89,90,91,92,93]. Capryl aldehyde and methyl-*n*-nonyl ketone obtained from *Houttuyniu cordata* directly inhibit the RT enzyme [66]. Calanolides A (**3**) and B (**4**) [89] have been obtained from the plant *Calophyllum inophyllum.* The introduction of bulky groups has been shown to be essential at the C-4 position to enhance anti-HIV activity. The stereochemistry of the C-12 hydroxyl (*R* or *S* configured) is not, however, as critical for activity. Methyl groups at the C-10 and C-11positions were also shown to be required for activity. Hydrogen bond acceptors at C-12 were also shown to be responsible for the activity, both in calanolides and inophyllums. In vitro assay results revealed that (+)-Calanolide-A inhibits RT in two diverse template primer systems. The action of (+)-Calanolide-A is possible due to the bi-bi prearranged mechanism of RT. Calanolide is at least partially competitive about dNTP binding. Structure-activity-relationships and important key findings of Calanolides are shown in Figure 4. 

Some naphthoquinones, e.g., Michellamines A, B and C, isolated from the plant *Ancistrocladus korupensis*, exhibited inhibitory activity against the HIV-RT enzyme [94]. The compound Acetogenin protolichensterinic acid is a RT inhibitory agent, obtained from *Cetraria islandica* [95]. The compounds Mallotochromene (**5**) and Mallotojaponin (**6**) (Figure 5) from *Mallotus japonicas* have shown strong inhibition against HIV-RT [96]. Nigranoic acid, from *Schisandra lancifolia*, acted effectively on the RT of HIV [97]. A few more examples of plants showing HIV-RT inhibitory properties are given in Table 1.

The compounds obtained from different plants, showing anti-HIV RT activity show the presence of certain pharmacophores which are essential for the therapeutic activity. These pharmacophoresinclude; coumarin, chromone, indole moiety and steroidal nucleous in the compounds, e.g., suksdorfin (**7**) [90], salaspermic acid (**8**) [99], cucurbitacin F (**9**) [104], batulinic acid (**10**) [107], baicalin (**11**) [109], buchenavianine (**12**) [110], thujone (**13**) [117], hyatellaquinone (**14**) [126], isodehyroluffariellolide (**15**) [127], homofascaplysin-C (**16**) [127], toxiusol (**17**) [128] reperesented in Figure 5.

#### 2.1.3. Plants Exhibiting Integrase Inhibition

The insertion of HIV DNA into the DNA of the host cell is generally catalyzed by the integrase enzyme of HIV. The reaction proceeds in two phases; the first phase is the processing phase and the second phase includes strand transfer [133]. Various active components have been separated from the plant *Dioscorea bulbifera* and exhibited several therapeutic properties such as: anticancer, antibacterial, analgesic, and antidiabetic [134,135,136,137,138]. Chaniad et al. isolated seven different compoundsfrom *D. bulbifera* which showed anti-HIV properties [45]. These include; allantoin (**18**), 5,7,4′-trihydroxy-2-styrylchromone,2,4,3′,5′-tetrahydroxybibenzyl, quercetin-3-*O*-β-D-galacto-pyranoside, 2,4,6,7-tetrahydroxy-9,10-dihydrophenanthrene, quercetin-3-*O*-β-D-glucopyranoside (**19**), and myricetin (**20**) (Figure 6). The results indicate that compound **20** had the best binding affinity within the active site of the integrase enzyme, forming strong interactions with amino acids. Moreover, significant activity is due to the presence of the galloyl, catechol, and sugar moieties which are responsible for the potential actions. In another study, Panthong et al. revealed that *Albizia procera* is a medicinal plant that has been used in antiretroviral therapy [139,140]. Catechin (**21**), suramin and protocatechuic acid (**22**) were shown to be the components identified from the plant extract and were considered to act on the integrase enzyme of HIV, thus prohibiting viral replication [139]. Compound **21** interacted with Thr66, Gly148, and Glu152 in the core domain of the enzyme, whereas compound **22** interacted with Thr66, His67, Glu152, Asn155, and Lys159. Some ribosome-inactivating proteins are considered to act on the integrase enzyme [141]. It was observed that compound **20**, having a galloyl moiety, possessed the most potent activity due to its strong binding with amino acids of the integrase enzyme. In compound **19**, the catechol group was partly responsible for the activity. Since compound **19** contains sugar moiety as well, which increases the solubility of the molecule, this enhances its activity. A ribosome-inactivating protein (RIP) named MAP30, which has been extracted from *Momordica charantia*, has been reported to act against HIV and cancer [142,143]. Zhao et al. discovered another RIP trichosanthin, from the roots of *Trichosanthes kirilowii*, which showed inhibitory activity against the integrase enzyme [144]. A variety of plant RIPs including agrostin, saporin, *R*-momorcharin, gelonin, α-momorchain, trichosanthin and luffin have also exhibited an inhibitory effect on HIV replication [145].

#### 2.1.4. Plants Containing Protease Inhibitors

Protease is a viral enzyme that acts at the last step of replication of the virus. It causes the breakdown of long polypeptides and proteins into the small functional proteins that are generally infectious [146,147,148,149]. Hence, protease is another target for the antiretroviral therapy and by inhibiting this enzyme the viral replication can be prohibited. Mostly, drugs act on this enzyme preferentially [147,149,150]. From the *Camellia japonica* pericarp, the plant components camelliatannins A, F and H have been reported that exhibited potent anti-HIV PR inhibitory properties [148]. Several Korean therapeutic plants, e.g., *Viburnum furcatum*, *IIex cornuta*, *Berberis amurensis*, *Lonicera japonica*, *Chloranthus glaber*, *Geranium nepalense*, *Lindera sericea*, *Wistaria floribunda*, *Smilax china*, *Hibiscus hamabo*, *Lingustrum lucidum*, *Zanthoxylum piperitum*, *Styrax obassia*, *Viola mandshurica*, *Schisandra nigra*, and *Cocculus trilobus* have also been reported potent activities against protease [71]. From the plant stems of *Stauntonia obovatifoliola*, various components that act against HIV protease have been identified, e.g., lupenone (**23**) [151], 3-*O*-acetyloleanolic acid (**24**) [152], resinone (**25**) [153], lupeol (**26**) [154] and mesenbryanthemoidgenic acid (**27**) (Figure 7) [155]. Moreover, the therapeutic compounds like oleanolic acid (**28**), dihydromyricetin, epigallocatechin gallate, myricetin [156,157] and epiafzelechin [158] have been extracted from the wood of *Xanthoceras sorbifolia* and have the potential for the treatment of AIDS [68,156]. 

#### 2.1.5. Plants Containing Immunomodulators

Immunomodulators are the agents that stimulate the cellular and humoral immune system against any pathogenic infection [159,160,161]. The dendritic cells of the immune system act as antigen representing cells and move along with antigens into the lymph nodes from the tissues. They represent the antigen to the T cells and the T cells then initiate an immune response. T cells stimulate the B cells for the production of antibodies that bind with the antigen and the T cells activate killer T cells which attack the pathogens [162]. There are several classes of naturally occurring compounds that exhibit immunomodulatory properties, e.g., alkaloids, tannins, terpenoids, coumarins, glycosides, flavonoids, polysaccharides, lignans, etc [163,164]. Among alkaloids, berberine (**29**) from *Hydrasti Canadensis* [159,165], sinomenine (**30**) from *Sinomenium acutum* [159,166], piperine (**31**) from *Piper longum* [159,167], and tetrandrine from *Stephania tetrandra* [168] have shown immunomodulatory properties in HIV. Among glycosides, aucubin from *Plantago major* [169], isorhamnetin-3-*O*-glucoside from *Urtica dioica* [170], and mangiferin from *Mangifera indica* [171] have exhibited immune-stimulatory properties in HIV. Among phenols, ellagic acid (**32**) from *Punica granatum* [172], curcumin from *Curcuma longa* [173], ferulic acid (**33**), vanilic acid (**34**) (Figure 8) [159] were shown to be immunostimulators in HIV. Chlorogenic acid from *Plantago major* [169], also expressed effective immunomodulatory potential in AIDS [159]. Within tannins, chebulagic acid and corilagin from *Terminalia chebula* [174] and punicalagin [175] acted as immunomodulatory agents. Among flavonoids, centaurein from *Bidens pilosa* [176] and apigenin 7-*O*-β-D-neohesperidoside, orientin, vitexin and apigenin 7-*O*-β-D-galactoside from *Jatropha curcas* [177] have exhibited the effective immunomodulatory action against HIV. From saponins, asiaticoside obtained from *Centella asiatica* [178] and glycyrrhizin from the roots of *Glycyrrhiza glabra* [179] have shown significant immunomodulatory activities.

#### 2.1.6. Plants with Antioxidant Potential

In AIDS, many reactive oxygen species (ROS) have been produced due to the alteration in the levels of antioxidant enzymes [180]. This further leads to the damage of DNA and lipid peroxidation [181]. ROS can also stimulate the nuclear factor kappa B (NF-κB factor) which helps in the transcription of HIV and thus promote its replication [182]. Antioxidants are agents that reduce the levels of ROS and protect cellular DNA. *N*-Acetylcysteine is reported to acts as an antioxidant and inturn used in themanagement of HIV infection [183]. Various other antioxidants like selenium, lipoic acid, vitamin C, β-carotene and vitamin E have been utilized for the same purpose [184,185]. The antioxidants Cyanidin-3-glucoside (**35**) and peonidin (**36**), which are obtained from blackberries, have also been shown to slow down AIDS infection (Figure 9) [159,186].

### 2.2. Classification of Plants According to Their Secondary Metabolites

Secondary metabolites are the main active compounds in plants that are mainly responsible for therapeutic effects. They are generally obtained from the primary metabolites such as carbohydrates, proteins, amino acids, etc. [187,188,189,190]. Plant-based secondary metabolites mainly include alkaloids, glycosides, coumarins, terpenoids, lignans, tannins and flavonoids, etc. [191,192,193].

#### 2.2.1. Alkaloids

Alkaloids are the basic nitrogen-containing secondary metabolites in plants, active against many pathogens, including HIV. Buchapine is a quinolone alkaloid obtained from *Eodia roxburghiana*, which has shown activity against HIV [194]. From the roots of *Tripterygium hypoglaucum*, various alkaloidal compounds have been isolated, e.g., hypoglaumine B, triptonine A (**37**) and B [159], which exhibited anti-HIV potential and have potential for antiretroviral therapy [195]. Nitidine is another alkaloid that was isolated from plant roots of *Toddalia asiatica*, and has shown efficacy against HIV [196]. From the plant *Symplocos setchuensis*, the alkaloid harman (**38**) and another compound matairesinoside (**39**) were isolated andshowed potential for antiretroviral therapy due to their anti-HIV potential. Compound **39** acts on the viral replication enzymes, thus inhibiting HIV replication [197]. Another aromatic alkaloid polycitone A, from marine source *Polycitor* sp., exhibited potential activity against the reverse transcriptase of HIV. Hence, it efficiently inhibits HIV replication. Several other marine sponges have acted against the virus as well as other bacterial diseases [198]. The alkaloid 1-methoxy canthionone was reported from *Leitneria floridana*, and exhibited anti-HIV property [199]. Papaverine was obtained from *Papaver sominiferum*, and inhibited HIV replication [39]. Norisoboldine and corydine are two alkaloids obtained from the leaves of *Croton echinocarpus*, showing anti-HIV activity [200]. Table 2 summarizes plant-based alkaloids possessing anti-HIV activity.

Several alkaloidal compounds, e.g., michellamine A (**40**) [201], siamenol (**41**) [203], decarine (**42**) [208], reticuline (**43**), norcoclaurine (**44**) [209], indole-3-carboxylic acid (**45**) [211], lycorine (**46**) [212], homolycorine (**47**) [212], cycleanine (**48**), 6-acetonyldihydrochelerythrine (**49**) [214] and hernandonine (**50**) [217] have revealed significant HIV inhibitory potential (Figure 10 and Figure 11).

#### 2.2.2. Terpenoids

Terpenoids are the secondary metabolites that are derived from the isoprene unit (C_5_H_8_). Terpenoids are the most abundant plant-based secondary metabolites and several compounds from this class have been derived from plants and found useful for their therapeutic potential [210,219,220]. Examples of terpenoids that have exhibited inhibition of HIV replication include betulinic acid, oleanolic acid, and platanic acid from *Syzigium claviflorum* leaves [221]. Celasdin B (**51**), is a triterpene from *Celastrus hindsii* (Celastraceae), reported inhibiting the HIV replication [222]. Prostratin from *Homalanthus nutans* (Euphorbiaceae) has also expressed significant anti-HIV activities [223]. From the stem bark of *Garcinia speciosa*, some anti-HIV therapeutic constituents have been isolated viz; garcisaterpenes A and C and theprotostanes. These compounds have been found to inhibit the activity of HIV reverse transcriptase and thus stop the HIV life cycle [224]. Maslinic acid (**52**), a terpenoid compound obtained from *Geum japonicum* also acts against the HIV protease enzyme [225]. From the stems and roots of plant *Kadsura lancilimba*, another triterpene lancilactone C (**53**) has been isolated whch is used to restrict the viral replication [226]. Oleanolic acid is the main terpenoid isolated from many plant species including *Xanthoceras sorbifolia* (Sapindaceae). The compound is known to inhibit HIV replication and play an important role in the treatment of AIDS [227]. Suberosol (**54**) is a lanostane type triterpenoid from the leaves of *Polyalthia suberosa* (Annonaceae) known toact through the same mechanism [228]. Another phorbol diester from *Croton tiglium* (Euphorbiaceae), 12-*O*-tetradecanoylphorbol-13-acetate, exhibited anti-HIV activity [223]. A Brazilian alga isolated from *Dictyota pfaffii*, from which an active diterpene component 8,10,18-trihydroxy-2,6-dolabelladiene has been isolated and has shown inhibitory activity of HIV reverse transcriptase [229,230]. A butenolide triterpene known as 3-*epi*-litsenolide has been articulated significant anti-HIV activity and was extracted from *Litsea verticilla* [231]. The alga *Dictyota menstrualis* is an important source for various diterpenes that exhibit HIV reverse transcriptase inhibition potential [232]. From the roots and rhizomes of plant *Clausena excavate*, a limonoid terpene named clausenolide-1-ethyl ether has shown potential for antiretroviral therapy [233]. Glycyrrhizin from the *Glycyrrhiza glabra* roots is another saponin terpenoid that showed anti-HIV activity by inhibiting the viral life cycle [234]. Oleanolic acid is a potent anti-HIV compound and is widely distributed in various plants including the leaves of *Rosa woodsii*, the leaves of *Syzygium claviflorum*, the aerial parts of *Ternstromia gymnanthera*, and the whole plants *Hyptis capitata* and *Phoradendron juniperinum* [227]. 12-Deoxyphorbol-13-phenylacetate, a phorbol ester from *Euphorbia poissonii*, has been reported for possessing anti-tumour activity and recently, it has potential for antiretroviral therapy because of its anti-HIV activity [235]. Pedilstatin [13-*O*-acetyl-12-*O*-(2′-*Z*-4′-Eoctadienoyl)-4-α-deoxyphorbol] is another phorbol ester from *Pedilanthus* sp., possessing anticancer and anti-HIV properties [236]. Some other plant species containing terpenoid based compounds with efficient anti-HIV activity have been summarized in Table 3.

Many of the terpenoid compounds, e.g., melliferone (**55**), whoseanti-HIV potentialhas been evaluated inanti-HIV assays towards T cell line H9, and compared with the positive control AZT has been shown in Figure 12 and Figure 13. Melliferone exhibited an IC_50_ value of 0.205 µg/mL [49], moronic acid (**56**), [49], ribenone (**57**) [237], germanicol (**58**) [243], nivadiol (**59**) [243], wikstroelide M (**60**) [252], shinjulactone B (**61**) [256], ganoderic acid D (**62**) [266], ganoderiol F (**63**) [267] gedunin (**64**) [276] and 1-α–hydroxy-1,2-dihydrogedunin (**65**) [276] also exhibited anti-HIV activities.

#### 2.2.3. Flavonoids

Flavonoids are well-known phytoconstituents reported to exhibit several antiviral and antioxidant properties [278]. Flavonoids like quercetin 3-*O*-(2-galloyl)-*L*-arbinopyranose and gallate ester from *Acer okamotoanum* (Aceraceae), exhibited significant activity against integrase of HIV [279]. Xanthohumol (**66**), an important flavonoid from *Humulus lupulus*, has shown anti-HIV activity [280]. The flavonoid moiety (4*H*-chromen-4-one) is known to be mainly responsible for the therapeutic activity, while glycosidic portion attached to the flavonoid enhances the solubility of the compounds and thus boosts its therapeutic activity. Two flavonoids 6,8-diprenylkaempferol and 6,8-diprenylaromadendrin isolated from *Monotes africanus* have expressed potential activity against the AIDS virus [281]. Another anti-HIV biflavonoid named wikstrol B (**67**) (Figure 14) has been isolated from *Wikstroemia indica* (Thymelaeaceae) roots [282]. Baicalin is a flavonoid compound that inhibits HIV replication and is derived from *Scutellaria baicalensis* [282]. From the twigs and leaves of the medicinal plant *Rhus succedanea,* various anti-HIV flavonoids (robustaflavone, biflavonoids, and hinokiflavone) have been reported to act on the polymerase of the reverse transcriptase of HIV-1 [283,284]. 2-methoxy-3-methyl-4,6-dihydroxy-5-(3′-hydroxy)-cinnamoylbenzaldehyde, a chalcone flavonoid that has been extracted from *Desmos* sp. roots and exhibited strong activity against HIV-1 [285]. Another chalcone, Hydroxypanduratin A, from the rhizomes of *Boesenbergia pandurata* depicted its action on the HIV protease enzyme [286].

Several naturally obtained flavonoids, e.g., chrysin, epigallocatechin gallate (**68**) and quercetin (**69**) have been reported to show potent inhibitory activities against the replication of HIV [276,277]. The flavonoids Thalassiolin A, B and C from the grass *Thalassia testudinum* acted against HIV integrase, which inturn inhibited the life cycle of HIV-1. Thalassiolin A was found to be the most potent compound which inhibits the terminal cleavage [287,288,289]. Some biflavonoids, e.g., 2″,3″-dihydroochnaflavone 7″-*O*-methylether and ochnaflavone 7″-*O*-methyl ether from *Ochna integerrima*, have shown moderate to weak anti-HIV activities [290,291]. Taxifolin (**70**), also known as dihydroquercetin, is mostly found the stems of *Juglans mandshurica*, and expressed strong inhibitory activity on the reverse transcriptase enzyme of HIV and thus plays a role in the prevention of HIV replication [292]. From *Chrysanthemum morifolium* flowers, two important flavonoids apigenin-7-*O*-β-D-(4′-caffeoyl)glucuronide and glucuronide have been isolated, which exhibited significant activity against the integrase of HIV-1 [293]. *Mentha longifolia* is another plant whose methanolic extracts are used for the isolation of several therapeutic flavonoids those were found to be active through the same mechanism [294]. Compound **70** was demonstrated to inhibitthe activity of HIV-1 replication. Several other flavonoids such as flemiphyllin, formosanatin C (**71**), euchretin I (**72**) and quercetin are reported to inhibit the HIV replication and obtained from the alcoholic extracts of *Euchresta formosana* [295]. Many important flavonoids such as epicatechin-3-*O*-gallate and epicatechin have extracted from *Detarium microcarpum*, have shown anti-HIV potential [296]. 4′-methylepigallocatechin-3′-*O*-β–glucopyranoside, and 4′-methylepigallocatechin-5-*O*-β-gluco-pyranoside from *Maytenus senegalensis* shown anti-HIV potential [297]. Kaempferol (**73**) (Figure 14), a tetrahydroxyflavonol was isolated from *Rosa damascene and* showed inhibitory activity on the protease enzyme [298,299].

#### 2.2.4. Coumarins

Calanolides are a group of coumarins that act as non-nucleoside reverse transcriptase inhibitors and are derived from plants of the genus *Calophyllum* (Clusiaceae) [300]. The coumarin (+)-Calanolide A has already been subjected to in vivo studies and up to phase II clinical trials in healthy, HIV-negative subjects. These studies revealthat (+)-calanolide A has a favourable safety profile in humans as well as in animals [301,302], while calanolide B alongwith its derivative known as 7,8-dihydrocalanolide B from the plant *Calophyllum lanigerum*, showedsignificant anti-HIV potential based on cytopathogenic results of HIV on the cells of the host [300]. Another coumarin named suksdorfin (**74**) [303,304] isolated from the fruits of *Lomatium suksdorfii* belonging to the family Apiaceae, which has expressed inhibitory property on the HIV replication [303]. The compounds Cordatolides A and B from *Calophyllum cordato-oblongum*, were similar in structureto the Calanolides and were found to inhibit the replication of the HIV [300]. The coumarin skeleton is essential for anti-HIV activity (Figure 15). Other coumarins like heraclenol (**75**) and heraclenin (**76**) exhibit IC_50_ value of 20.1 µg/mL against H9 lymphocytes, while imperatorin (**77**) from the roots of *Ferula sumbul* falls under the same therapeutic category [305]. Bulky groups at the C-4 are also required to retain the anti-HIV activity, which is present in the prototype of a molecule like (+)-Calanolide-A. (+)-Calanolide-A is the most potent compound when compared with Cordatolide A (less active and devoid of the bulky group at the C-4 position). Several furanocoumarins (e.g., bergapten (**78**) and psolaren) from the roots of *Prangos tschimganica*, have exhibited significant activities against the HIV virus [306]. Mesuol (**79**) is another coumarin (from the category 4-phenylcoumarin) reported to inhibit the replication of HIV-1through the prohibition of the reverse transcription and phosphorylation of HIV [307]. A semisynthetic derivative of calanolide (known as oxocalanolide) was alsoreported to act efficiently against HIV [308]. Various furanocoumarins (e.g., imperatorin, xanthotoxin and xanthotoxol) have been extracted from the *Aegle marmelos* fruits [121,122]. The stem, roots, fruits, leaves, seeds and bark of the *A. marmelous* showed variable antiviral effects and have played an important role in Ayurvedic medicine. Imperatorin (**77**) is reported to exhibit about 60% inhibition of HIV-RT. The absence of a prenyl group resulted in the observed weak activity. This is exemplified in the cases of other furanocoumarins xanthotoxin (**80**) and xanthotoxol (**81**), shown in Figure 15 [309,310].

#### 2.2.5. Proteins

Proteins are the amino acid-containing plant components that usually contain ribosome-inactivating proteins as well as lectins [311]. A plant protein called MAP30 from *Momordica charantia* is known to possess anticancer potential along with anti-HIV properties [312]. Various plant ribosome-inactivating proteins have been identified for their anti-HIV activities. Trichosanthin is a ribosome-inactivating protein isolated from *Trichosanthes kirilowii* that has shown anti-HIV activity [313]. Various plant ribosome-inactivating proteins have been identified for their anti-HIV activities, e.g., an anti-HIV ribosome-inactivating protein balsamin has been extracted from *Momordica balsamina* [314]. Pf-gp6 is another protein reported from *Perilla frutescens* which has exhibited an inhibitory action on HIV replication [315]. Some ribosome-inactivating proteins known as Pokeweed antiviral proteins have been separated from a pokeweed plant (*Phytolacca americana*) and have expressed efficient anti-HIV activities [316]. A list of plant proteins has been given in Table 4, along with their botanical sources.

#### 2.2.6. Tannins

Tannins are mainly categorized into gallotannins and ellagitannins. While gallotannins are hydrolysable and contain gallic acid polyesters ellagitannins are non-hydrolyzable, so-called condensed tannins conatining hexahydroxydiphenic acids, i.e., flavan-3-ol (proanthocyanidins) moieties [345,346]. Corilagin (**82**) and Geraniin (**83**) (Figure 16), from roots of *Phyllanthus amarus*, are two ellagitannins that possess anti-HIV activities [347]. Besides, a proanthocyanidin compound from the plant *Cupressus sempervirens*, exerted anti-HIV properties [348]. Catechins, the polyphenols that are obtained from green tea, and theaflavins (e.g., compound **84**) isolated from black tea, possess antiviral activity. Theaflavins and their derivatives are potent inhibitors of HIV replication [349]. Compounds **82** and **83** blocked the interaction of HIV-1 gp120 with its primary cellular receptor CD4. Besides, the observed results showed that compound 83 exhibited inhibitory effects on HIV, not only in vitro but also in vivo. Compound **84** inhibited HIV-1 entry into target cells by blocking the HIV-1 envelope glycoprotein-mediated membrane fusion. The ability of this compound to block the formation of the gp41 six-helix bundle was determined using Fluorescence native polyacrylamide gel electrophoresis, while detection of the binding of gp120 to CD4 was done by ELISA. Molecular docking analyses suggested that compound **84** may bind with to the highly conserved hydrophobic pocket on the surface of the central trimeric coiled-coil of gp41.

#### 2.2.7. Lignans

Several extensive reports on plant-based lignans which have shown strong activities against viral diseases, including AIDS, exist [350]. Several lignans like anolignan A (**85**) and anolignan B, alongwith dibenzylbutadiene lignans have been isolated from *Anogeissus acuminate* and have exhibited significant activity against HIV [351]. From *Phyllanthus myrtifolius* (Euphorbiaceae), phyllamyricin D (**86**) and phyllamyricin F (**87**) (Figure 17) were isolated and shown to possess inhibitory activity against the HIV-RT enzyme [352]. The benzoaryl moiety was proven to be essential for the anti-HIV activity of lignans. This group is responsible for inhibiting HIV replication. Gomisin is another example of lignan isolated from *Kadsura interior* and showed potent inhibitory activity against the RT enzyme of HIV [353]. From *Arnebia euchroma*, some caffeic acid isomers have been evaluated but have only expressed weak activities against HIV replication [354]. The compound 2-hydroxy-2 (3′,4′-dihydroxyphenyl)-methyl-3-(3″,4″-dimethoxyphenyl) methyl γ–butyrolactone is a dibenzylbutyrolactone type lignan from *Phenax angustifolius* with established anti-HIV activity [355]. From *Schisandra rubriflora* fruits, other dibenzocyclooctadiene type lignans (rubrisandrin A and rubrisandrin B) have been isolated having anti-HIV activities [356].

#### 2.2.8. Miscellaneous Plant-Based Anti-HIV Agents

Numerous plants have been evaluated for their anti-HIV activities and are being used in antiretroviral therapy for AIDS [1,2]. Various phenolic compounds isolated from plants such as *Quercus pedunculata, Terminalia horrida, Phyllanthus emblica* and *Rumex cyprius* have been identified for their anti-HIV activities [357,358]. From the leaves and twigs of plant *Strychnos vanprukii*, various betulinic acid derivatives, such as 3-β-*O*-*cis*-feruloylbetulinic acid (**88-B**), 3-β-*O*-*trans*-feruloylbetulinic acid (**88-A**), ursolic acid and 3-β-*O*-*trans*-coumaroylbetulinic acid (**89**) have exhibited potential against HIV [359]. Compounds **88-A**, **88-B** and **89** have been evaluated for anti-HIV activities against HOG.R5 cells in the anti-HIV assay. Compound **88-A** showed significant inhibition against HIV-1 replication. The *trans*-isomer (**88-A**) showed a more favourable activity when compared with the *cis*-isomer (**88-B**) (Figure 18). The compounds shown in Figure 18 exhibited significant potential against HIV due to the presence of the pharmacophore/heterocyclic moieties, such as chromone, indole, steroidal nucleus, benzodioxole, quinolizine, etc. These compounds also demonstrated various therapeutic properties, e.g., anti-inflammatory, anti-cancer, antiviral, antioxidant and immunomodulatory properties [360]. The constituents of *Cinnamomum zylanicum* bark have shown anti-inflammatory [361], anti-cancer, antiviral, antioxidant and immunomodulatory properties [362]. The ingenol compounds from *Euphorbia ingens* have exhibited anti-HIV activities [363], apart from their anti-inflammatory and immunomodulatory potentials [364,365]. *Oldenlandia affinis* is a medicinal plant from which various cyclotides have been isolated and tested for their activities against HIV [366,367]. *Plectranthus barbatus* has also shown diverse antiviral, antibacterial and antifungal properties along with antioxidant and anti-inflammatory effects [368,369]. From *Clausena excavate* some therapeutic constituents like *O*-methylmukonal (**90**), 3-formyl-2,7-dimethoxycarbazole, limonoids, and clausenidin have been reported for their anti-HIV properties [370,371]. Several antiviral components like tectorigenin, cytisine (**91**), formononetin, trifolirhizin (**92**), mattrine (**93**), blumenol A (**94**), pterocarpin (**95**), 30,40,5-trihydroxyisoflavone, euchretin and 5,7-dihydroxy-3-(2-hydroxy-4-methoxy-phenyl)-chromen-4-one have been isolated from *Euchresta formosana* and exhibited anti-HIV activities [372,373,374,375].

Extracts from *Alepidea amatymbica*, have shown efficient anti-HIV activities, as well as inhibitory effect on HIV replication [376]. Artemisinin from the plant *Artemisia annua*, has established antimalarial and anti-HIV activities [377]. Rosmarinic acid is a polyphenolic compound from the plant *Prunella vulgaris*, usedfor the treatment of isolated HIV [378]. From *Polygonum glabrum*, various bioactive constituents with antiretroviral activities have been reported, e.g., (-)-2-methoxy-2-butenolide-3-cinnamate, pinocembrin (**96**), 3-hydroxy-5-methoxystilbene (**97**), sitosterol-3-*O*-β-D-glucopyranoside, and pinocembrin-5-methyl ether [379]. Actein (**98**) from the rhizomes of *Cimicifuga racemosa*, possessed a significant activity against HIV [380]. Chrysoeriol from *Eurya ciliate* is known for its anti-HIV activity [381]. Several constituents such as demethylaristofolin E (**99**), aristofolin, denitroaristolochic acid, aristolochic acid, aristomanoside (**100**), *N-p*-coumaroyltyramine, *p*-hydroxybenzoic acid, etc. have been isolated for their anti-HIV potential from the stem bark of *Aristolochia manshuriensis* [382,383,384,385,386]. Malaferin A, from *Malania oleifera*, was also tested for its antiviral property [387]. Diptoindonesin D, Acuminatol (**101**), Shoreaphenol, Hopeahainol, and Vaticanol B from *Vatica mangachapoi* have shown positive effects in the management of antiretroviral therapy [388,389,390,391]. Cararosinol C and D, maackin and scirpusin B (**102**) from *Caragana rosea* have been evaluated for their anti-HIV effects [392]. Structures of some important constituents obtained from plants effective in HIV therapy are represented in Figure 18. A list of other plants having anti-HIV potential has been listed in Table 5.

## 3. Conclusions

Plants are known to exhibit a huge repertoire of bioactive metabolites [436]. A significant number of reports on the capability of natural compounds with potential as anti-HIV agents have appeared during the last few decades. This review article presents the rational approaches for the design of therapeutic potential candidates as anti-HIV agents. Even though there have been many extensive achievements in the field of HIV chemotherapy, there remains a great demand for novel lead compounds for anti-HIV drug discovery and drug development. Numerous plant species have been evaluated for their inhibitory activities on the essential HIV enzymes such as RT, protease, and integrase, which play an important role in HIV replication. Several secondary metabolites have been extracted from the various parts of plantsthat act as potent anti-HIV agents via different mechanisms of action. Therapeutically active compounds from plants can also aid as necessary leads for the discovery and development of novel and more potent compounds that can be derived synthetically. For instance, synthetic ingenol compounds have been derived based on naturally occurring compound Ingenol and a variety of synthetic derivatives have been evolved from the naturally occurring compound Artemisinin, which exhibits significant anti-HIV activitiesof potential scaffolds from them for the complete eradication of HIV/AIDS. A recent review has attempted to show themost successful medical therapeutics derived from natural products, including those studied in the field of HIV/AIDS [437]. Besides, computer-aided (virtual) [438] and large-scale in vitro screening [439] approaches have recently been carried out on natural compound libraries to identify natural products with anti-HIV properties. Novel therapeutic approaches have been attempted, including searching for new HIV-1 latency-reversing agents, i.e., compounds not only capable of HIV suppression but also eliminating HIV reservoirs [440,441].

## Figures and Tables

**Figure 1 molecules-25-02070-f001:**
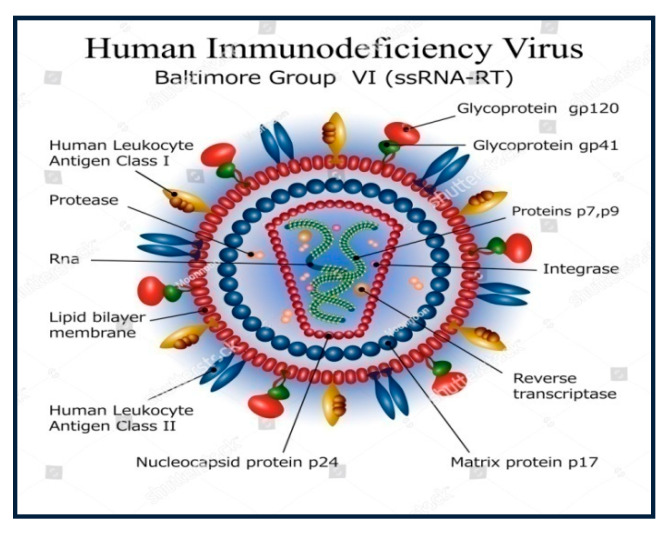
Structure of human immunodeficiency virus (HIV) virus [10]. Image was originally published within Open Access license.

**Figure 2 molecules-25-02070-f002:**
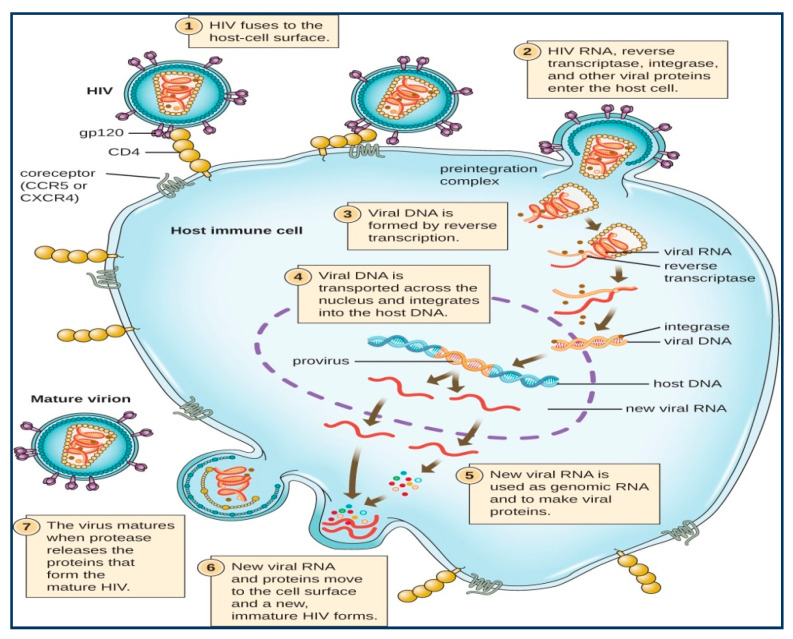
The HIV replication cycle [13]. Image was originally published within Open Access license.

**Figure 3 molecules-25-02070-f003:**
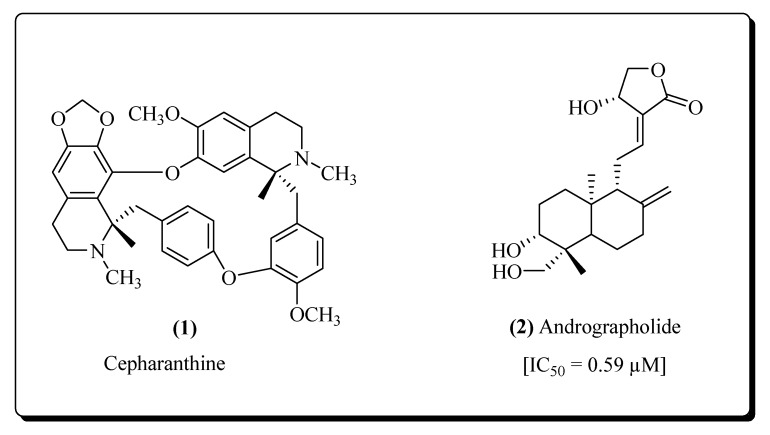
Structures of fusion inhibitors.

**Figure 4 molecules-25-02070-f004:**
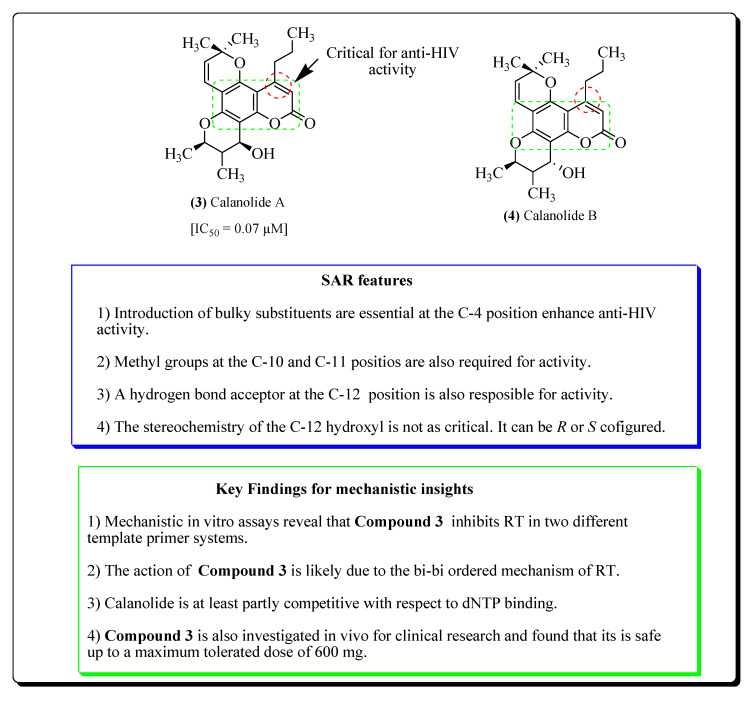
Structure-activity-relationships and important key findings of some potent reverse transcriptase (RT) inhibitors.

**Figure 5 molecules-25-02070-f005:**
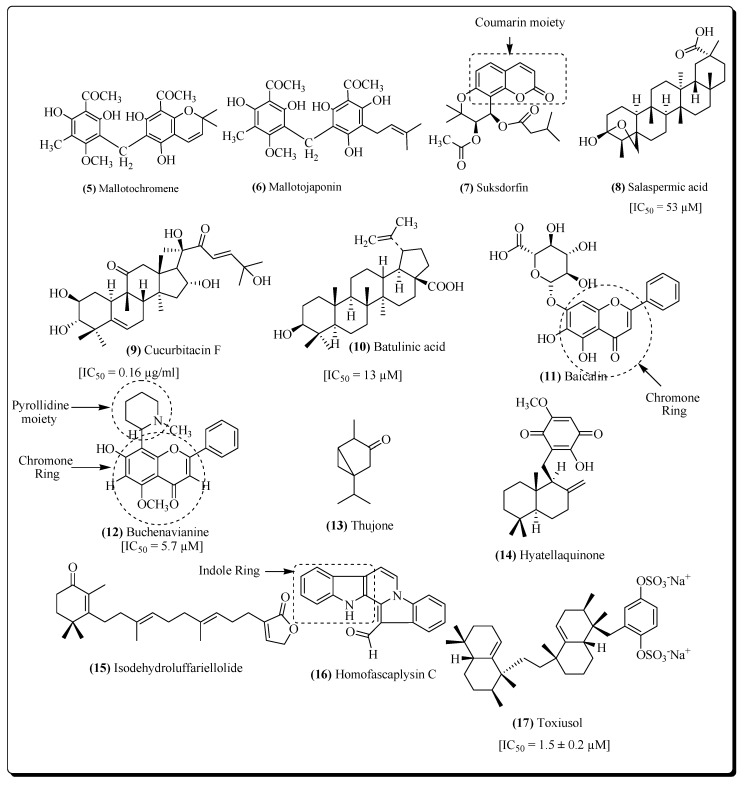
Structure of some potent reverse transcriptase inhibitors.

**Figure 6 molecules-25-02070-f006:**
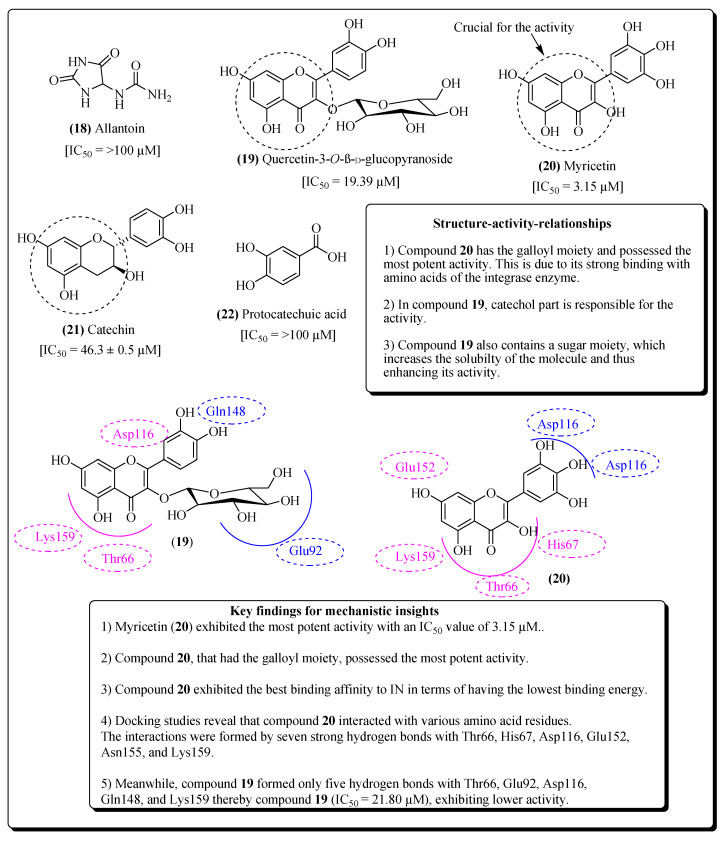
Structure-activity-relationships of naturally occurring integrase inhibitors.

**Figure 7 molecules-25-02070-f007:**
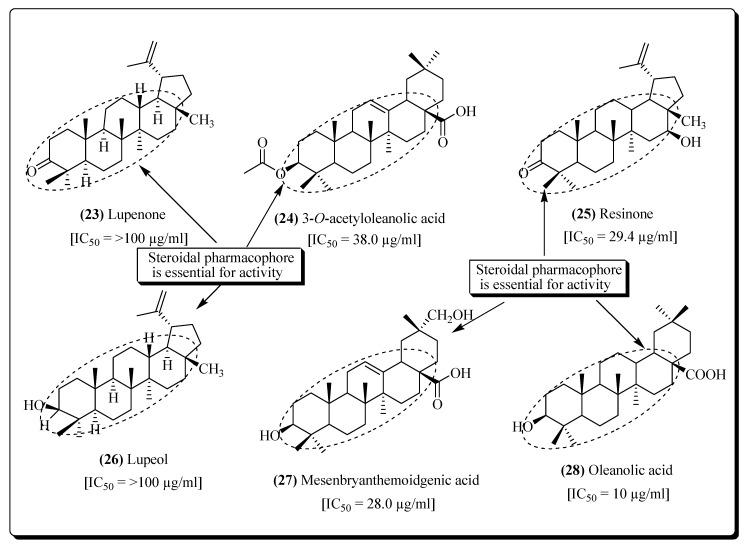
Compounds exhibiting protease inhibitory activity.

**Figure 8 molecules-25-02070-f008:**
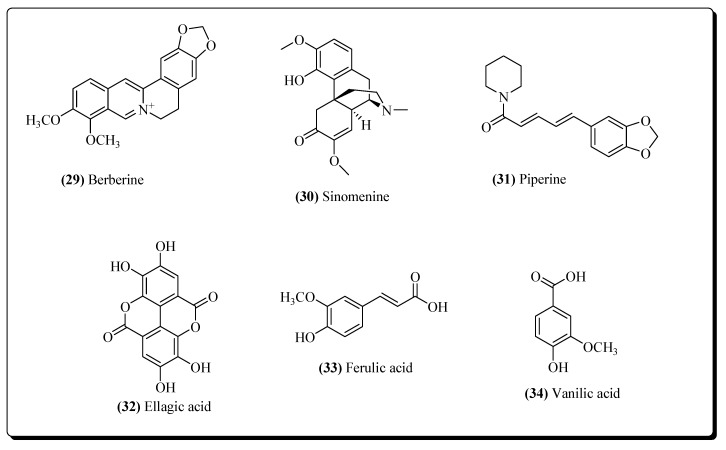
Plant-based Immunomodulators.

**Figure 9 molecules-25-02070-f009:**
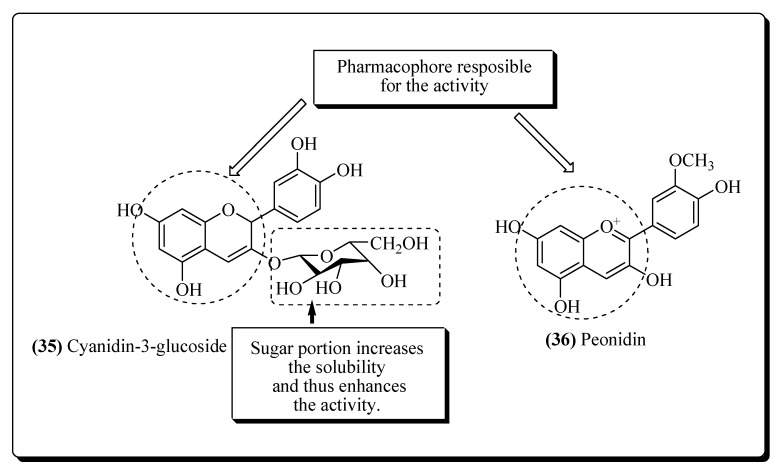
Plant-based antioxidants compounds possessing anti-HIV potential.

**Figure 10 molecules-25-02070-f010:**
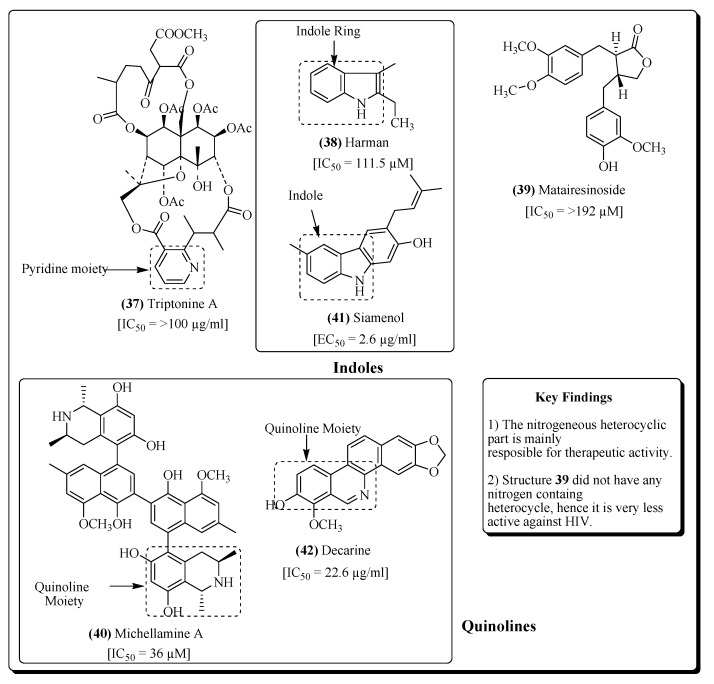
Alkaloidal compounds possessing anti-HIV activity.

**Figure 11 molecules-25-02070-f011:**
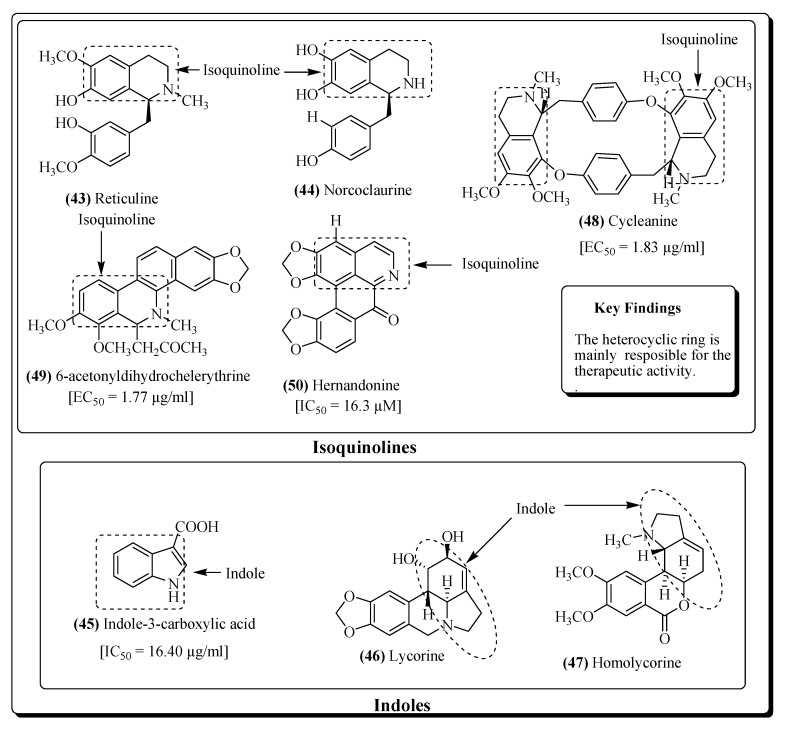
More alkaloidal compounds possessing anti-HIV activity.

**Figure 12 molecules-25-02070-f012:**
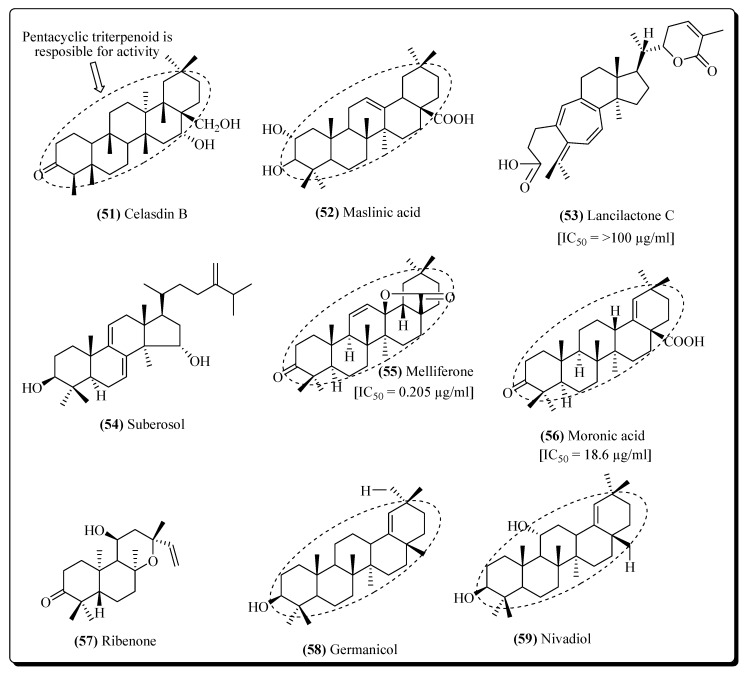
Potent terpenoids against HIV.

**Figure 13 molecules-25-02070-f013:**
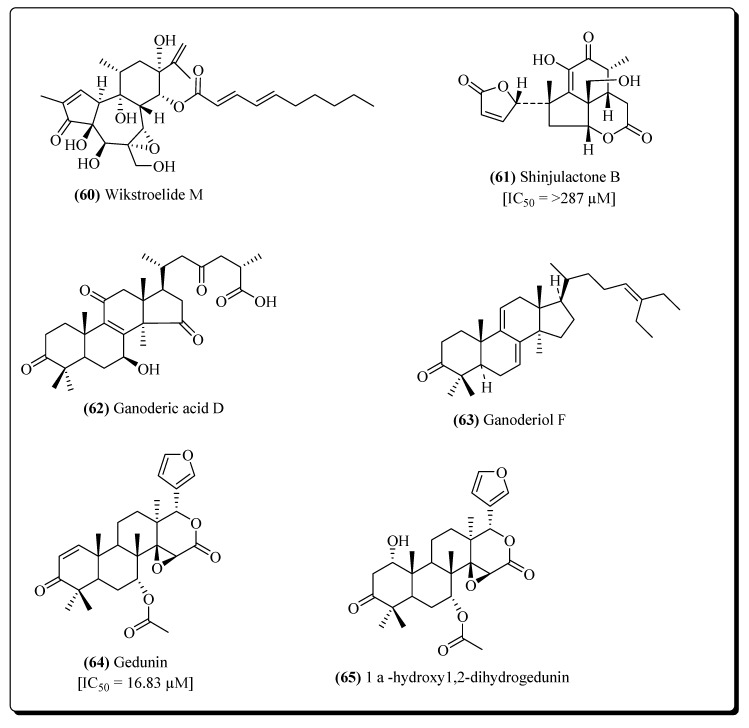
More potent terpenoids against HIV.

**Figure 14 molecules-25-02070-f014:**
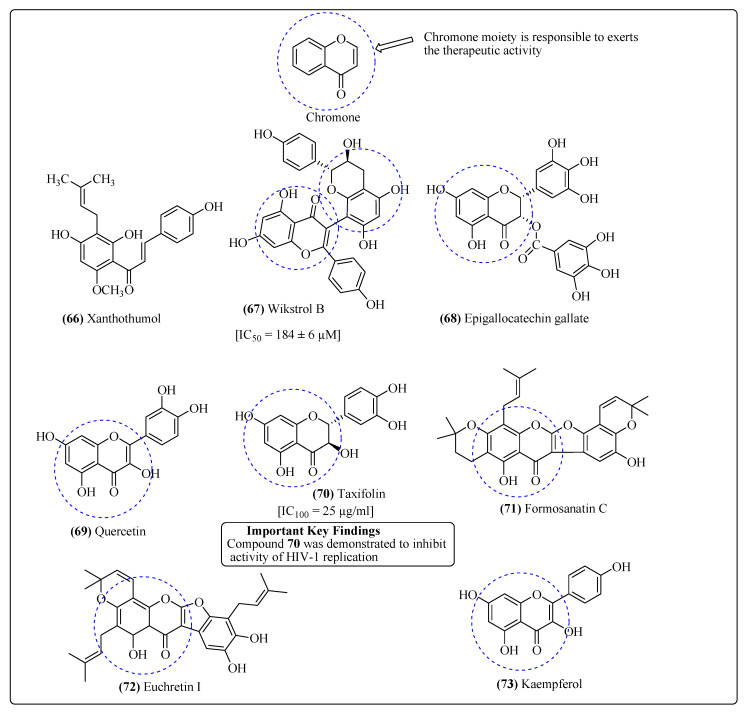
Flavanoids with anti-HIV properties.

**Figure 15 molecules-25-02070-f015:**
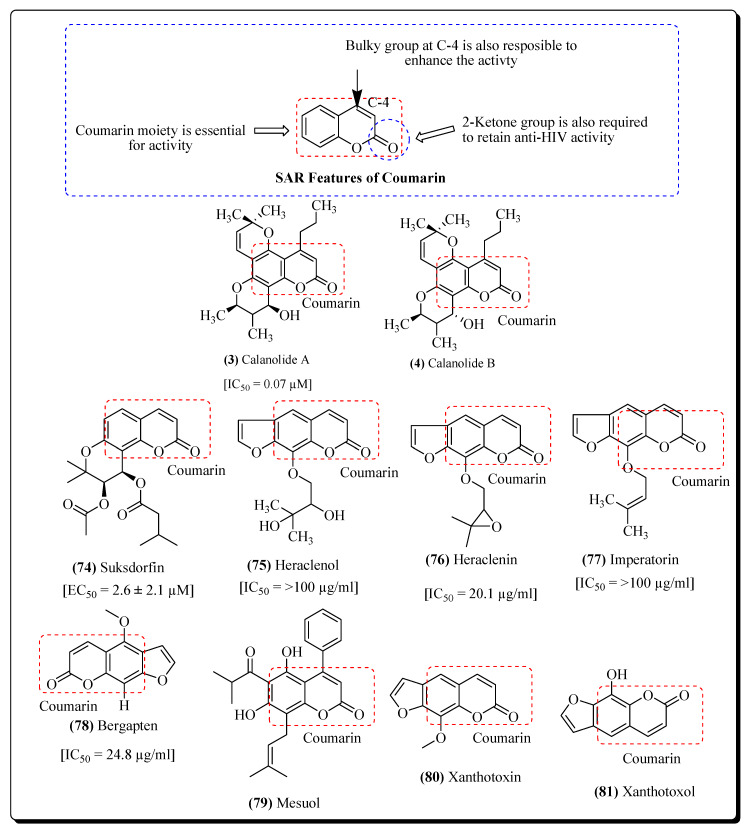
Coumarins with significant Anti-HIV potential.

**Figure 16 molecules-25-02070-f016:**
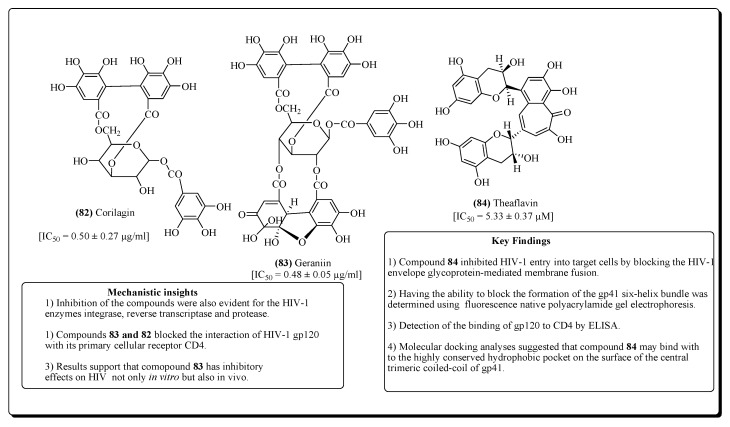
Tannins with anti-HIV properties.

**Figure 17 molecules-25-02070-f017:**
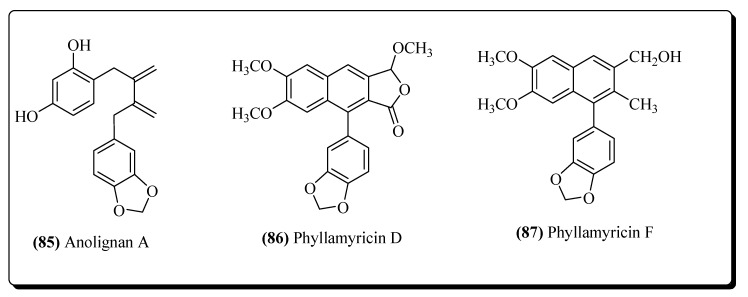
Lignans possessing anti-HIV activities.

**Figure 18 molecules-25-02070-f018:**
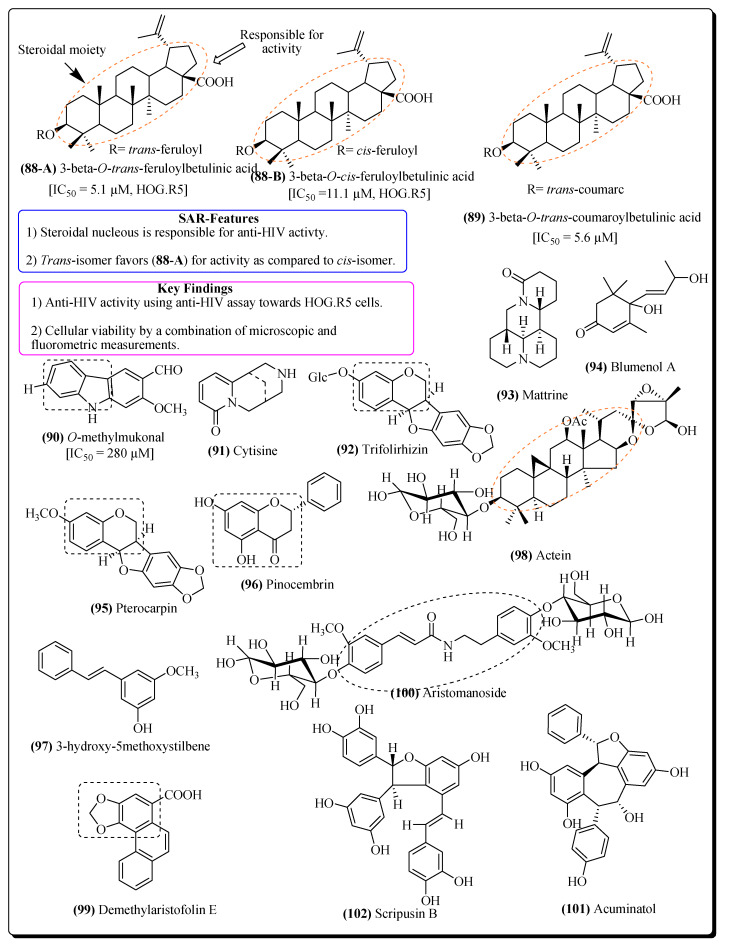
Other plant-based compounds with anti-HIV activities.

**Table 1 molecules-25-02070-t001:** Plant-based reverse transcriptase inhibitors.

Compound Class	Plant Species	Chemical Constituents	Reference
Terpenoid	*Excoecaria agallocha*	Phorbol	[98]
Terpenoid	*Trypterygium wilfordii*	Salaspermic acid	[99]
Terpenoid	*Euphorbia myrsinites*	15-*O*-acetyl-3-*O*-butanoyl-5-*O*-propionyl-7-*O*-nicotinoylmyrsinol	[100]
Terpenoid	*Polyalthia suberosa*	Suberosol	[101]
Terpenoid	*Andrographis paniculata*	Dehydroandrographolide succinic acid monoester	[102]
Terpenoid	*Glycyrrhiza radix*	Glycyrrhizin	[103]
Terpenoid	*Cowania Mexicana*	Cucurbitacin F	[104]
Terpenoid	*Tripterygium wilfordii*	Tripterifordin	[105]
Terpenoid	*Maprounea Africana*	1 β-hydroxymaprounic acid 3-*p*-hydroxybenzoate	[106]
Terpenoid	*Szigium claviforum*	Betulinic acid, platonic acid	[107]
Terpenoid	*Houttuynia cordata*	Lauryl aldehyde, capryl aldehyde	[108]
Flavonoid	*Chrysanthemum morifolium*	Acacetin-7-*O*-β-galactopyranoside	[92]
Flavonoid	*Scutellaria baicalensis*	Baicalin	[109]
Flavonoid	*Buchenavia capitata*	Buchenavianine	[110]
Flavonoid	*Kummerolvia striata*	Apigenin-7-*O*-β-D-glucopyranoside	[111]
Coumarin	*Calophyllum inophyllum*	Inophyllums	[112]
Coumarin	*Coriandrum sativum*	Coriandrin	[91]
Coumarin	*Lomatium suksdorfii*	Suksdorfin	[90]
Coumarin	*Aegle marmelous*	Imperatorin, xanthotoxol, xanthotoxin	[113,114]
Tannin	*Euphorbia jolkini*	Putranjivain A	[115]
Tannin	*Cornus officinalis*	Cornusin A	[116]
Tannin	*Mallotus repandus*	Repandusinic acid	[117]
Tannin	*Hyssop officinalis*	Caffeic acid	[118]
Polysaccharide	*Thuja occidentalis*	Thujone	[119]
Polysaccharide	*Prunella vulgar*	Sulfated polysaccharide	[120]
Polysaccharide	*Viola yedoensis*	Sulfonated polysaccharide	[121]
Xanthone	*Tripterospermum lenceolaum*	1,3,5,6-tetrahydroxyxanthone,	[122]
		3,4,5,6-tetrahydroxyxanthone	
Lignan	*Haplophyllum ptilostylum*	Ptilostin	[123]
Lignan	*Schisandra chinensis*	Gomisin J	[124]
Lignan	*Ipomoea cairica*	Arctigenin, trachelogenin	[125]
Marine origin	*Hyatella intestinalis*	Hyatellaquinone	[126]
Marine origin	*Fascaplysinopis reticulate*	Fascaplysin, isodehydroluffariellolide,	[127]
		Homofascaplysin C	
-	*Toxiclona toxius*	Toxiusol	[128]
-	*Plakortis* sp.	Plakinidine A	[129]
-	*Kelletia kelletii*	Kelletinin 1	[130]
-	*Buccinulum corneum*	Kelletinin A	[131]
-	*Maprounea Africana*	1β-hydroxyaleuritolic acid 3-*p*-hydroxybenzoate	[132]

**Table 2 molecules-25-02070-t002:** Alkaloidal compounds as anti-HIV agents.

Plant Species	Parts Used	Chemical Constituents	References
*Ancistrocladus korupensis*	Leaves	Michellamine A, B and C	[201]
*Stephania cepharantha*	Roots	Cepharantine	[202]
*Murraya siamensis*	Roots, leaves	Siamenol	[203]
*Clausena excavate*	Leaves	*O*-Methylmukonal, clauszoline and 3-formyl-2,7-dimethoxy-carbazole	[204]
*Drymaria diandra*	Leaves	Canthin-4-one drymaritin	[205]
*Glycosmis montana*	Twigs, leaves	(*E*)-3-(3-hydroxymethyl-2-butenyl)-7-(3-methyl-2-butenyl)-1*H*-indole	[206]
*Aniba* sp.	Stems	Anibamine	[207]
*Zanthoxylum ailanthoides*	Root bark	Decarine, *ϒ*-fagarine and tembamide	[208]
*Nelumbo nucifera*	Leaves	Coclaurine, norcoclaurine, reticuline	[209]
*Pericampylus glaucus*	Leaves	Norruffscine, 8-oxotetrahydro-palmatine	[210]
*Begonia nantoensis*	Rhizomes	Indole-3-carboxylic acid	[211]
*Leucojum vernum*	Bulbs	Lycorine, homolycorine	[212]
*Epinetrum villosum*	Root bark	Cycleanine	[213]
*Argemone mexicana*	Roots	6-acetonyldihydrochelerythrine, nuciferine	[214]
*Monanchora* sp.	Stems	Crambescidin 826, fromiamycalin and crambescidin 800	[215]
		Manadomanzamines A and B	[216]
*Acanthostrongylophora* sp.	-	Hernandonine, lindechunine,7-oxohernangerine and laurolistine	[217]
	Roots		
*Lindera chunii*			
*Artemisia caruifolia*	Stems		[218]

**Table 3 molecules-25-02070-t003:** Terpenoids act as Anti-HIV agents.

Plant Species	Parts Used	Chemical Constituents	References
*Excoecaria acerifolia*	Roots	Agallochin J, ribenone, angustanoic acid B	[237,238,239]
*Propolis*	Roots	Melliferone, moronic acid, betulonic acid	[49,240]
*Homalanthus nutans*	Leaves	Prostratin	[241,242]
*Cassine xylocarpa*	Stem	Germanicol, nivadiol	[243]
*Glycyrrhiza uralensis*	Roots	Galacturonic acid, xylose, uralsaponin C	[244]
*Daphne gnidium*	Aerial parts	Daphnetoxin, gniditrin, gnidicin	[245]
*Euphorbia micractina*	Roots	Lanthyrane diterpenoids	[246]
*Kaempferia pulchra*	Rhizomes	Kaempulchraol A, C, E	[247]
*Picrasama javanica*	Bark	Picrajavanicin A, javanicin B, picrasin A	[248]
*Schisandra lancifolia*	Leaves, stem	Lancifodilactone F	[249]
*Stellera chamaejasme*	Roots	Stelleralide D, gnidimacrin	[250]
*Lindera strychnifolia*	Roots	Lindenanolides E, G and F	[251]
*Daphne acutiloba*	Roots	Wikstroelide M	[252]
*Annona squamosa*	Leaves	16-β,17-dihydroxy-entkauran-19-oic acid	[253]
*Cimicifuga racemosa*	Rhizomes	Actein	[254]
*Schisandra sphaerandra*	Leaves	Nigranoic acid	[255]
*Allanthus altissima*	Roots	Shinjulactone B	[256]
*Panax ginseng*	Roots	Isodehydroprotopanaxatriol	[257]
*Garcinia hanburyi*	Stem, roots	3-acetoxyalphitolic acid, 2-acetoxyalphitolic acid	[258]
		8-methoxyingol-7,12-diacetate-3-phenylacetate	
*Euphorbia officinarum*	Leaves	Dihydrocucurbitacin F	[259]
		Forskolin, 1-deoxyforskolin	
*Hemsleya jinfushanensis*	Tubers	28-hydroxy-3-oxo-lup-20(29)-en-3-*O*-al	[260]
*Coleus forskohlii*	Roots	Betulonic acid	[261]
*Microtropis fokienensis*	Stem	25-hydroxy-3-oxoolean-12-en-28-oic acid	[262]
*Betula platyphylla*	Roots	Capilliposide B	[263]
*Amoora rohituka*	Stem bark	Ganoderic acid D	[264]
		Ganoderiol F	
*Lysimachia capillipes*	Roots	Impatienside A, bivittoside D	[265]
*Ganoderma lucidum*	Stem, Leaves	25-methoxyhispidol A	[266]
*Ganoderma amboinense*	Stem	23,24-dihydrocucurbitacin B	[267]
*Holothuria impatiens*	-	Dichapetalin A	[268]
*Poncirus trifoliate*	Fruits	Acutissimatriterpene A, B, E	[269]
*Trichosanthes kirilowii*	Roots	Celastrol	[270]
*Dichapetalum gelonioides*	Stem bark	3α,7α-dideacetylkhivorin	[271]
*Phyllanthus acutissima*	Aerial parts	Nimbolide	[272]
*Celastrus orbiculatus*	Bark	Gedunin, 1 α–hydroxy-1,2-dihydrogedunin	[273]
*Khaya senegalensis*	Roots	6α-tigloyloxychaparrinone	[274]
*Azadirachta indica*	Flowers		[275]
*Xylocarpus granatum*	Roots		[276]
*Ailanthus integrifolia*			[277]

**Table 4 molecules-25-02070-t004:** Proteins containing plants used in HIV.

Plant Species	Parts Used	Proteins	References
*Allium ascalonicum*	Bulbs	Ascalin	[317]
*Chrysanthemum coronarium*	Seeds	Chrysancorin	[318]
*Ginkgo biloba*	Seeds	Ginkbilobin	[319]
*Arachis hypogaea*	Seeds	Hypogin	[320]
*Lyophyllum shimeji*	Fruit bodies	Lyophyllin	[321]
*Panax quinquefolium*	Roots	Quinqueginsin	[322]
*Flammulina velutipes*	Fruit bodies	Velutin	[323]
*Tricholoma giganteum*	Fruit bodies	Laccase protein	[324]
*Castanea mollisima*	Seeds	Mollisin	[325]
*Treculia obovoidea*	Bark	Treculavirin	[326]
*Vigna sesquipedalis*	Seeds	Ground bean lectin	[327]
*Delandia unbellata*	Seeds	Delandin	[328]
*Dorstenia contrajerva*	Leaves	Contrajervin	[326]
*Vigna angularis*	Seeds	Angularin	[329]
*Castanopsis chinensis*	Seeds	Castanopsis thaumatin protein	[330]
*Vigna unguiculata*	Seeds	Cowpea α protein	[331]
*Phaseolus vulgaris*	Seeds	A homodimeric lectin	[332]
*Actinidia chinensis*	Fruits	Kiwi fruit thaumatin protein	[333]
*Lentinus edodes*	Fruit bodies	Lentin	[334]
*Allium tuberosum*	Shoots	A mannose-binding lectin	[335]
*Phaseolus vulgaris*	Seeds	Phasein A	[336]
*Lilium brownie*	Bulbs	Lilin	[337]
*Vicia faba*	Seeds	A trypsin-chymotrypsin	[338]
		Inhibitor peptide	
*Vigna unguiculata*	Seeds	Unguilin	[339]
*Panax notoginseng*	Roots	A xylanase	[340]
*Phaseolus vulgaris*	Seeds	Vulgin	[341]
*Cicer arietinum*	Seeds	Chickpea cyclophilin-like protein	[342]
		α–Basrubrin	
*Basella rubra*	Seeds	Rice bean peptide	[343]
*Delandia unbellata*	Seeds		[344]

**Table 5 molecules-25-02070-t005:** Assortments of other plant species have been given in Table 5.

Plant Species	Family	Parts Used	References
*Khaya grandifoliola*	Meliaceae	Leaves	[393]
*Diospyros mespiliformis*	Ebenaceae	Bark	[394]
*Alternanthera brasiliana*	Amaranthaceae	Roots	[395]
*Ricinus communis*	Euphorbiaceae	Leaves	[396]
*Butea monosperma*	Fabaceae	Roots	[397]
*Prosopis glandulosa*	Fabaceae	Leaves	[398]
*Sophora tonkinensis*	Fabaceae	Roots	[399]
*Gunnera magellanica*	Gunneraceae	Stem	[400]
*Swertia franchetiana*	Gentianaceae	Roots	[401]
*Curcuma longa*	Zingiberaceae	Rhizomes	[402]
*Stewartia koreana*	Theaceae	Leaves	[403]
*Cissus quadrangularis*	Vitaceae	Stems	[404]
*Withania somnifera*	Solanaceae	Roots	[405]
*Ailanthus altissima*	Simaroubaceae	Stem bark	[406]
*Toddalia asiatica*	Rutaceae	Roots	[407]
*Oldenlandia herbacea*	Rubiaceae	Roots	[408]
*Aloe vera*	Xanthorrhoeaceae	Leaves	[409]
*Urtica dioica*	Urticaceae	Rhizomes	[410]
*Rheum tanguticum*	Polygonaceae	Leaves	[411]
*Saccharum officinarum*	Poaceae	Stems	[412]
*Ochna integerrima*	Ochnaceae	Leaves	[413]
*Nelumbo nucifera*	Nelumbonaceae	Leaves	[414]
*Aglaia lawii*	Meliaceae	Leaves	[415]
*Fritillaria cirrhosa*	Liliaceae	Rhizomes	[416]
*Magnolia biondii*	Magnoliaceae	Flower buds	[417]
*Lythrum salicaria*	Lythraceae	Leaves	[418]
*Reseda lutea*	Resedaceae	Whole plant	[419]
*Hypericum perforatum*	Hypericaceae	Leaves	[420]
*Trigonostemon thyrsoideus*	Euphorbiaceae	Stems	[421]
*Hemsleya endecaphylla*	Cucurbitaceae	Tubers	[422]
*Garcinia kingaensis*	Clusiaceae	Stem bark	[423]
*Woodwardia unigemmata*	Blechnaceae	Rhizomes	[424]
*Berberis holstii*	Berberidaceae	Roots, leaves	[425]
*Foeniculum vulgare*	Apiaceae	Fruits	[406]
*Alepidea amatymbica*	Apiaceae	Roots	[426]
*Stachytarpheta jamaicensis*	Verbenaceae	Whole plant	[427]
*Schisandra sphaerandra*	Schisandraceae	Stems	[428]
*Alpinia galangal*	Zingiberaceae	Roots	[429]
*Zanthoxylum chalybeum*	Rutaceae	Root bark	[430]
*Berchemia berchemiifolia*	Rhamnaceae	Bark	[431]
*Scoparia dulcis*	Plantaginaceae	Leaves	[432]
*Phyllanthus myrtifolius*	Phyllanthaceae	Fruits	[433]
*Arundina graminifolia*	Orchidaceae	Whole plant	[434]
*Ximenia Americana*	Olacaceae	Stem bark	[435]

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
