# Peer review of "Structure-Activity-Relationship and Mechanistic Insights for Anti-HIV Natural Products"

_molecules, 2020, doi:10.3390/molecules25092070_

Round 1

Reviewer 1 Report

My comments were appropriately addressed in the revision. I would however suggest extensive English editing to section 1. HIV pathogenesis (section 1.1)  subtitle is also misleading since the content describes the HIV structure. The section that describes the HIV lifecycle too needs to be refocused with appropriate reference to receptors and not statements like "the virus invade into the body cells through various binding receptors that are depicting on the top of the macrophages, T lymphocytes, dendritic cells and monocytes"_these receptors have names.

Author Response

My comments were appropriately addressed in the revision. I would however suggest extensive
English editing to section 1. HIV pathogenesis (section 1.1) subtitle is also misleading since the
content describes the HIV structure.
Author's response:
The English has been thoroughly checked and the section heading has been changed.
The section that describes the HIV lifecycle too needs to be refocused with appropriate reference
to receptors and not statements like "the virus invade into the body cells through various binding
receptors that are depicting on the top of the macrophages, T lymphocytes, dendritic cells and
monocytes"_these receptors have names.
Author's response:
Thanks for the comment. This has been addressed.

Reviewer 2 Report

Journal: Molecules
Manuscript ID: molecules-770175
Type of manuscript: Review
Title: Structure-activity relationship and mechanistic insights for anti-HIV natural products

Authors: Ramanpreet Kaur, Pooja Sharma, Girish Gupta, Fidele Ntie-Kang, Dinesh Kumar

Since several plants have displayed significant anti-HIV activity, there is an urgent demand to search anti-HIV agents from plant species. For this purpose, the present review article summarizes the most potent plant-derived anti-HIV compounds, with the mechanisms of action and IC50 values discussed. However, the organization of the manuscript is not logic, and language is not perfect to express clearly the authors' opinions or conclusions. The authors should combine “Plants according to their mechanism of action” and “Plants according to the activity of secondary metabolites” to discuss plant-derived anti-HIV agents. Thus, they may re-organize the manuscript, following the sequence alkaloids, coumarins, flavonoids including xanthones, lignans, polysaccharides, proteins, tannins, terpenoids, marine origins, and miscellaneous plant secondary metabolites. Information about the host plants and mechanisms of action, including inhibition of fusion, reverse transcriptase, integrase, and protease and immunomodulation and antioxidant activity, of each class of compounds listed in the sequence could be included in the paragraph that they summarized each of these types of natural products. A reference may be interested for the authors to follow, which is Shriwas, P., et al. Phytotherapy Research 2019, DOI: 10.1002/ptr.6587, as included in the comments. In addition, English should be improved a lot. Therefore, this review article is recommended to be reconsidered after major revision as a potential review for Molecules.

Author Response

Reviewer#2:

Open Review

English language and style

(x) Extensive editing of English language and style required 
( ) Moderate English changes required 
( ) English language and style are fine/minor spell check required 
( ) I don't feel qualified to judge about the English language and style 

Is the work a significant contribution to the field?

Is the work well organized and comprehensively described?

Is the work scientifically sound and not misleading?

Are there appropriate and adequate references to related and previous work?

Is the English used correct and readable?

Comments and Suggestions for Authors

Journal: Molecules
Manuscript ID: molecules-770175
Type of manuscript: Review
Title: Structure-activity relationship and mechanistic insights for anti-HIV natural products

Authors: Ramanpreet Kaur, Pooja Sharma, Girish Gupta, Fidele Ntie-Kang, Dinesh Kumar

Since several plants have displayed significant anti-HIV activity, there is an urgent demand to search anti-HIV agents from plant species. For this purpose, the present review article summarizes the most potent plant-derived anti-HIV compounds, with the mechanisms of action and IC50 values discussed. However, the organization of the manuscript is not logic, and language is not perfect to express clearly the authors' opinions or conclusions.

Author's response:

The English has been thoroughly checked and some text reorganized.

The authors should combine “Plants according to their mechanism of action” and “Plants according to the activity of secondary metabolites” to discuss plant-derived anti-HIV agents. Thus, they may re-organize the manuscript, following the sequence alkaloids, coumarins, flavonoids including xanthones, lignans, polysaccharides, proteins, tannins, terpenoids, marine origins, and miscellaneous plant secondary metabolites.

Author's response:

Since the manuscript targets a special issue of the journal that focuses on SAR of natural products, our aim was to both address the SMs by both mode of actions, e.g. RT inbitors, protease inhibitors, etc, then approach the discussion by compound classes, which partly explains the length of the manuscript.

Information about the host plants and mechanisms of action, including inhibition of fusion, reverse transcriptase, integrase, and protease and immunomodulation and antioxidant activity, of each class of compounds listed in the sequence could be included in the paragraph that they summarized each of these types of natural products.

Author's response:

The approach suggested by the reviewer is appreciated. However, that would require twisting our entire text, since this was not the author's originally thought approach. It must be mentioned that this is the second submission after comments from several referees.

A reference may be interested for the authors to follow, which is Shriwas, P., et al. Phytotherapy Research 2019, DOI: 10.1002/ptr.6587, as included in the comments.

Author's response:

The suggested reference has been incorporated in the updated manuscript.

In addition, English should be improved a lot. Therefore, this review article is recommended to be reconsidered after major revision as a potential review for Molecules.

Author's response:

The English has been thoroughly checked and some text reorganized.

Submission Date

26 March 2020

Date of this review

28 Mar 2020 21:07:16

Reviewer 3 Report

This manuscript is very intersting and well-conceived. It should be accepted for publication after some minor revisions aimed at simplifing some figures and the main findings. I'd like to suggest the following revisions:

Line 43 WHO – explain the acronym eve if it is well-known

Line 61 Reverse  transcriptase enzyme – please introduce the acronym RT

Line 112 change “diseae” with “disease”

Lines 116-119 The sentence “Many types of combination approaches such as use of nucleoside 117 reverse transcriptase inhibitors, fusion inhibitors, non-nucleoside reverse transcriptase inhibitors, 118 integrase inhibitors, protease inhibitors, together with immunomodulators have been prescribed to 119 achieve a proficient therapeutic response” should be properly accompanied by references

Line 163 change “ sourece” with “source”

Line 221 change “litreraure” with “literature”

Line 229 correct the sentence “It can be R or S cofigured”

Fig. 5 please use monoletter code for aminoacids. SAR and key findings summarized in this figure should be removed from the black box and briefly discussed within the main text of the review.

Fig. 9 this figure could be modified dividing the compounds into 3 groups based on the indole-containing the quinolone and isoquinolines main core

Move the important key findings from fig.11 and 13 to the main text or include a brief sentence along with the manuscript to underline the key findings. Simplify also fig 15 or divide it into two figures based on the chemical structure of the compounds.

Author Response

Reviewer#3:

Open Review

English language and style

( ) Extensive editing of English language and style required 
( ) Moderate English changes required 
(x) English language and style are fine/minor spell check required 
( ) I don't feel qualified to judge about the English language and style 

Is the work a significant contribution to the field?

Is the work well organized and comprehensively described?

Is the work scientifically sound and not misleading?

Are there appropriate and adequate references to related and previous work?

Is the English used correct and readable?

Comments and Suggestions for Authors

This manuscript is very intersting and well-conceived. It should be accepted for publication after some minor revisions aimed at simplifing some figures and the main findings. I'd like to suggest the following revisions:

Line 43 WHO – explain the acronym eve if it is well-known

Author's response:

Thanks for this remark. This has been corrected.

Line 61 Reverse  transcriptase enzyme – please introduce the acronym RT

Author's response:

Thanks for this remark. This has been corrected.

Line 112 change “diseae” with “disease”

Author's response:

Thanks for this remark. This has been corrected.

Lines 116-119 The sentence “Many types of combination approaches such as use of nucleoside 117 reverse transcriptase inhibitors, fusion inhibitors, non-nucleoside reverse transcriptase inhibitors, 118 integrase inhibitors, protease inhibitors, together with immunomodulators have been prescribed to 119 achieve a proficient therapeutic response” should be properly accompanied by references

 Author's response:

Thanks for this remark. The appropriate references have now been included.

Line 163 change “ sourece” with “source”

Author's response:

Thanks for this remark. This has been corrected.

Line 221 change “litreraure” with “literature”

Author's response:

Thanks for this remark. This has been corrected.

Line 229 correct the sentence “It can be R or S cofigured”

Author's response:

Thanks for this remark. This has been corrected.

Fig. 5 please use monoletter code for aminoacids. SAR and key findings summarized in this figure should be removed from the black box and briefly discussed within the main text of the review.

Author's response:

Thanks for this remark. The authors made the choice to use the three letter codes throughout the text.  has been corrected. The SAR has now been incorporated in the text (lines 255-258).

Fig. 9 this figure could be modified dividing the compounds into 3 groups based on the indole-containing the quinolone and isoquinolines main core

Author's response:

Thanks for this remark. The authors have made the change in the updated manuscript.

Move the important key findings from fig.11 and 13 to the main text or include a brief sentence along with the manuscript to underline the key findings.

Author's response:

Thanks for this remark. The authors have made the change in the updated manuscript.

Simplify also fig 15 or divide it into two figures based on the chemical structure of the compounds.

Author's response:

Thanks for this remark. The authors were, however, unable to split these, since so many classes of chemical structures are present in this figure..

In general the entire manuscript was read through to improve the grammar and double-checked using the grammarly software. The reference list was checked again to be sure all references were correctly written.

Submission Date

26 March 2020

Date of this review

01 Apr 2020 17:27:40

Round 2

Reviewer 2 Report

Journal: Molecules
Manuscript ID: molecules-770175
Type of manuscript: Review
Title: Structure-activity relationship and mechanistic insights for anti-HIV natural products

Authors: Ramanpreet Kaur, Pooja Sharma, Girish Gupta, Fidele Ntie-Kang, Dinesh Kumar

The present review article summarizes the most potent plant-derived anti-HIV compounds, with the mechanisms of action and IC50 values discussed. However, the organization of the manuscript is not logic, and language is not perfect to express clearly the authors' opinions or conclusions. Therefore, this review article is recommended to be reconsidered after major revision as a potential review for Molecules.

  1. The authors should combine “Plants according to their mechanism of action” and “Plants according to the activity of secondary metabolites” to discuss plant-derived anti-HIV agents.
  2. Check Figure 4 and update other Figures. It looks like that the authors listed two Figure 4.
  3. Check 2.1 and following 1.5.1. It seems not consistent.
  4. Include Figure 3 in Figures 9 and 10, Figure 4 (first) in Figure 12, Figure 4 (second) in Figures 9, 10, 11, and 15, Figure 5 in Figures 9, 11, and 15, Figure 6 in Figure 10, Figure 7 in Figures 9, 13, and 15, Figure 8 in Figure 11.
  5. Re-organize the manuscript, following the sequence alkaloids, coumarins, flavonoids including xanthones, lignans, polysaccharides, proteins, tannins, terpenoids, marine origins, and miscellaneous plant secondary metabolites.
  6. Include the information about the host plants and mechanisms of action, including inhibition of fusion, reverse transcriptase, integrase, and protease and immunomodulation and antioxidant activity in each class of compounds discussed.
  7. Tables 1–5 should be updated following these changes. Are these tables necessary if they have already been discussed in the text?

Author Response

The present review article summarizes the most potent plant-derived anti-HIV compounds, with the mechanisms of action and IC50 values discussed. However, the organization of the manuscript is not logic, and language is not perfect to express clearly the authors' opinions or conclusions. Therefore, this review article is recommended to be reconsidered after major revision as a potential review for Molecules.

  1. The authors should combine “Plants according to their mechanism of action” and “Plants according to the activity of secondary metabolites” to discuss plant-derived anti-HIV agents.

Authors' response:

The suggestion is welcome but could not be implemented in the current manuscript at this advanced stage.

  1. Check Figure 4 and update other Figures. It looks like that the authors listed two Figure 4.

Authors' response:

The figure numbering has now been reordered chronologically

  1. Check 2.1 and the following 1.5.1. It seems not consistent.

Authors' response:

The numbering of the headings and sub-headings has been redone.

  1. Include Figure 3 in Figures 9 and 10, Figure 4 (first) in Figure 12, Figure 4 (second) in Figures 9, 10, 11, and 15, Figure 5 in Figures 9, 11, and 15, Figure 6 in Figure 10, Figure 7 in Figures 9, 13, and 15, Figure 8 in Figure 11.

Authors' response:

This would be completely confusing and would disrupt the entire layout of the manuscript.

  1. Re-organize the manuscript, following the sequence alkaloids, coumarins, flavonoids including xanthones, lignans, polysaccharides, proteins, tannins, terpenoids, marine origins, and miscellaneous plant secondary metabolites.

Authors' response:

The authors understand that the intention of the reviewer is to present an alphabetical order. However, this suggestion would mess up the entire manuscript and reference numbering.

  1. Include the information about the host plants and mechanisms of action, including inhibition of fusion, reverse transcriptase, integrase, and protease and immunomodulation and antioxidant activity in each class of compounds discussed.

Authors' response:

This information has been included in the text (where available). Do you suggest that we also include in the tables?

  1. Tables 1–5 should be updated following these changes. Are these tables necessary if they have already been discussed in the text?

Authors' response:

The contents of the tables have been maintained, as the order of the text has not changed.

This manuscript is a resubmission of an earlier submission. The following is a list of the peer review reports and author responses from that submission.

Round 1

Reviewer 1 Report

This interesting review includes a lot of information about AIDS and the treatments against it, using recent references and including a long list of bioactive compounds. However, I have some comments about some parts of the text.

As far as I know, no statement about the copyright of the images contained in this manuscript and the authorship was included. The image on figure 1 is published the Webpage www.123rf.com by the user moonnoon, specifically, it can be found on this link: www.123rf.com/photo_26111982_schéma-de-la-structure-des-particules-de-virus-hiv.html

However, instead of the original author, the reference mentioned is number 8, a manuscript in Nature Reviews Microbiology. I think the authors should mention if they have the consent of the original author to publish this figure in an Open Access Journal.

Same case with the image in figure 2, is not found in reference 6 (Annual Review of Pathology: Mechanisms of Disease), it can be found with the exact same text in https://www.niaid.nih.gov/diseases-conditions/hiv-replication-cycle

In this case, nonetheless, this image is included in several sites with minor modifications in shape and color.

Some parts of the text contain sentences where the wording is a little confusing, so I recommend the authors to revise the text. For example, lines 99-100: it is not clear for me by reading the text is the authors mean if either the Western Blotting or the alternatives at ref. 18 are less time consuming. If the alternatives are less time consuming after the ‘which’ ‘are’ instead of ‘is’ should be used. At line 496: ‘lignans are the secondary metabolites of the plants’, I do not understand what the authors mean with that. Line 514: the way the sentence is written implies that Q. pedunculata and the rest of the plants are phenolic compounds. Needs to be rephrased.

I found similar examples of unconnected sentences or wrong punctuation along all the text so I would strongly recommend the authors to revise it.

The structure of andrographolide in figure 3 should follow the same format as the other molecules. The stereochemistry of one of the hydroxyls was drawn with a slash style and the other hydroxyl and rest of substituents with hyphens.

The structure of quercetin-3-O-β-D-glucopyranoside should be corrected, the sugar residue should have the correct stereochemistry (also in figure 8, cyanidin-3-glucoside). Also the name is wrong in both the figure and the text. The oxygen atom should be written in capital italics (also the nitrogen in line 307 for N-acetylcysteine, hydrogen in table 2, oxygen again in line 367, etc), the B should be a beta (in the figure) and the D should be written in ‘small caps’. Please, apply this to other molecule names containing sugar residues.

The hydrogen in the fourth fused six-member ring of lupenone and lupeol in figure 6 does not need to be drawn; it is not in a chiral center.

E and Z, when referring to the configuration of double bonds should be in italics (table 2, 3rd column, 6th row). ‘p’ (para) when using it to identify the positions of the functional groups in an aromatic ring should be in italics (line 545).

In figure 9 contains several mistakes: the structures of reticuline and norcoclaurine lack the stereochemistry and one of the aromatic rings in cycleanine is deformed.

Structures in figure 10 also need corrections. There is no coherence between the styles, some molecules show CH3 either as CH3 or in the shortened form (as ‘sticks’), for example celasdin B and maslinic acid. Maslinic acid shows extra hydrogens in the first ring at the right that are not needed (not a chiral center). Lancilactone C drawing is wrong, both CH3 in the double bond should be in the same plane and the angle with the ring in wrong, the CH2 connecting with the lactone also need to be redrawn correctly. The six-member ring of melliferone is deformed. In general, in molecules that contain two CH3 at the same carbon the angles are not correct and different in each molecule. In wikstroelide M the angles of the carbonyl groups are wrong.

In figure 12, the epoxide in heraclenin contains wrong angles.

In figure 14, phyllamyricin D and F structures show extra hydrogens that are not needed.

According to reference 367, the name for ‘(2)-2-methoxy-2-butenolide-3-cinnamate’ should be corrected to ‘(-)-2-methoxy-2-butenolide-3-cinnamate’, the (2) has no sense and is a mistake in other references from the literature. Reference 367 first author name has to be corrected to Said.

In figure 15, the sugar rest of actein should have the stereochemistry of the ring indicated. The hydrogen in the double bond of scripusin B has the wrong angle.

There also some minor mistakes: ‘natral plants’ (line 208), ‘prooved’ (line 271), ‘replicativr’ (line 329), ‘3formyl’ (table 2, 3rd column, 4th row), ‘canthin4-one’ (same as previous, 5th row), ‘an another’ (line 366), ‘islotaed’ (line 421), ‘7-Oβ-D-(4’caffeoyl)‘ (line 430), ‘3’-Oβ-‘ (line 438), ‘-Dglucopyranoside’ (line 540).

Author Response

Comments and Suggestions for Authors

This interesting review includes a lot of information about AIDS and the treatments against it, using recent references and including a long list of bioactive compounds. However, I have some comments about some parts of the text.

Author's response: We appreciate this comment.

All the corrections have been incorporated in the final version of manuscript and are highlighted in yellow color

As far as I know, no statement about the copyright of the images contained in this manuscript and the authorship was included. The image on figure 1 is published the Webpage www.123rf.com by the user moonnoon, specifically, it can be found on this link: www.123rf.com/photo_26111982_schéma-de-la-structure-des-particules-de-virus-hiv.html

Authors' response:

A comment has been included under the figure caption.

However, instead of the original author, the reference mentioned is number 8, a manuscript in Nature Reviews Microbiology. I think the authors should mention if they have the consent of the original author to publish this figure in an Open Access Journal.

Authors' response:

A comment has been included under the figure caption.

Same case with the image in figure 2, is not found in reference 6 (Annual Review of Pathology: Mechanisms of Disease), it can be found with the exact same text in https://www.niaid.nih.gov/diseases-conditions/hiv-replication-cycle

Authors' response:

A comment has been included under the figure caption.

In this case, nonetheless, this image is included in several sites with minor modifications in shape and color.

Authors' response:

A comment has been included under the figure caption.

Some parts of the text contain sentences where the wording is a little confusing, so I recommend the authors to revise the text. For example, lines 99-100: it is not clear for me by reading the text is the authors mean if either the Western Blotting or the alternatives at ref. 18 are less time consuming. If the alternatives are less time consuming after the ‘which’ ‘are’ instead of ‘is’ should be used. At line 496: ‘lignans are the secondary metabolites of the plants’, I do not understand what the authors mean with that. Line 514: the way the sentence is written implies that Q. pedunculata and the rest of the plants are phenolic compounds. Needs to be rephrased.

Authors' response:

We appreciate this comment. We have included these suggested changes on the updated revised manuscript. ‘lignans are the secondary metabolites of the plants means that they belongs to secondary metabolites and these secondary metabolites of plants are responsible for therapeutic effects’.

I found similar examples of unconnected sentences or wrong punctuation along all the text so I would strongly recommend the authors to revise it.

Authors' response:

The final version of the manuscript has been carefully revised in final version of manuscript.

The structure of andrographolide in figure 3 should follow the same format as the other molecules. The stereochemistry of one of the hydroxyls was drawn with a slash style and the other hydroxyl and rest of substituents with hyphens.

Authors' response:

We appreciate this comment. We have included these suggested changes on the updated revised manuscript. The structure of andrographolide in figure 3 have been corrected as recommended by the honorable reviewer.

The structure of quercetin-3-O-β-D-glucopyranoside should be corrected, the sugar residue should have the correct stereochemistry (also in figure 8, cyanidin-3-glucoside). Also the name is wrong in both the figure and the text. The oxygen atom should be written in capital italics (also the nitrogen in line 307 for N-acetylcysteine, hydrogen in table 2, oxygen again in line 367, etc), the B should be a beta (in the figure) and the D should be written in ‘small caps’. Please, apply this to other molecule names containing sugar residues.α

Authors' response:

We appreciate this comment. We have included these suggested changes on the updated revised manuscript.

The hydrogen in the fourth fused six-member ring of lupenone and lupeol in figure 6 does not need to be drawn; it is not in a chiral center.

Authors' response:

We appreciate this comment. We have included these suggested changes on the updated revised manuscript. We agree with the reviewer and as suggested, In figure 6 hydrogen in the fourth fused six member ring of lupenone and lupeol have been removed in updated revised manuscript.

E and Z, when referring to the configuration of double bonds should be in italics (table 2, 3rd column, 6th row). ‘p’ (para) when using it to identify the positions of the functional groups in an aromatic ring should be in italics (line 545).

Authors' response:

We appreciate this comment. We have included these suggested changes on the updated revised manuscript.

In figure 9 contains several mistakes: the structures of reticuline and norcoclaurine lack the stereochemistry and one of the aromatic rings in cycleanine is deformed.

Authors' response:

We appreciate this comment. We have included these suggested changes on the updated revised manuscript.

Structures in figure 10 also need corrections. There is no coherence between the styles, some molecules show CH3 either as CH3 or in the shortened form (as ‘sticks’), for example celasdin B and maslinic acid. Maslinic acid shows extra hydrogens in the first ring at the right that are not needed (not a chiral center). Lancilactone C drawing is wrong, both CH3 in the double bond should be in the same plane and the angle with the ring in wrong, the CH2 connecting with the lactone also need to be redrawn correctly. The six-member ring of melliferone is deformed. In general, in molecules that contain two CH3 at the same carbon the angles are not correct and different in each molecule. In wikstroelide M the angles of the carbonyl groups are wrong.

Authors' response:

We appreciate this comment. We have included these suggested changes on the updated revised manuscript.

In figure 12, the epoxide in heraclenin contains wrong angles.

Authors' response:

We appreciate this comment. We have included these suggested changes on the updated revised manuscript.

In figure 14, phyllamyricin D and F structures show extra hydrogens that are not needed.

Authors' response:

We appreciate this comment. We have included these suggested changes on the updated revised manuscript. As suggested by honorable reviewer In figure 14, phyllamyricin D and F structures extra hydrogens have been removed.

According to reference 367, the name for ‘(2)-2-methoxy-2-butenolide-3-cinnamate’ should be corrected to ‘(-)-2-methoxy-2-butenolide-3-cinnamate’, the (2) has no sense and is a mistake in other references from the literature. Reference 367 first author name has to be corrected to Said.

Authors' response:

We appreciate this comment. We have included these suggested changes on the updated revised manuscript.

In figure 15, the sugar rest of actein should have the stereochemistry of the ring indicated. The hydrogen in the double bond of scripusin B has the wrong angle.

Authors' response:

We appreciate this comment. We have included these suggested changes on the updated revised manuscript.

There also some minor mistakes: ‘natral plants’ (line 208), ‘prooved’ (line 271), ‘replicativr’ (line 329), ‘3formyl’ (table 2, 3rd column, 4th row), ‘canthin4-one’ (same as previous, 5th row), ‘an another’ (line 366), ‘islotaed’ (line 421), ‘7-Oβ-D-(4’caffeoyl)‘ (line 430), ‘3’-Oβ-‘ (line 438), ‘-Dglucopyranoside’ (line 540).

Authors' response:

We appreciate this comment. We agree with the reviewer and as suggested, we have included these suggested changes in the updated revised manuscript.

Reviewer 2 Report

This is a nearly exhaustive review of published literature on natural product derived compounds with anti-HIV activity. In general, the sections of the review assessing various classes of compounds are way too descriptive when they should be critical and synthetic. One would expect an in-depth overview of SAR and mechanistic insights, but it never really comes out in the review. Instead, we get many pages that only list numerous compounds with no description of mechanistic work, controls and assays used to determine antiviral activities of the natural compounds. HIV pathogenesis is naively written with numerous scientific errors. It would be much more exciting if the authors could give us an overview of what is known, methods used to perform a literature search of the compounds described, any evidence of clinical trials on such compounds, translational challenges to be overcome in moving forward natural products with anti-HIV activity, instead of telling the reader on how existing antiretroviral therapy are bad when they have saved millions of lives. An important aspect to consider is are there any reports that perform direct comparison between herbal medicines and existing anti-HIV medicines to demonstrate considerable benefits or improvements? Most sections of the review too are difficult to follow due to numerous grammatical errors. Overall, this review needs to be rewritten and focused

Author Response

Is the English used correct and readable?

Comments and Suggestions for Authors

This is a nearly exhaustive review of published literature on natural product derived compounds with anti-HIV activity.

Authors' response:

We appreciate this comment. But we enlighten that this is a comprehensive and systematic review on the natural plant derived compound with anti-HIV activity. And this review is very helpful for the researchers who are working on this area of research.

In general, the sections of the review assessing various classes of compounds are way too descriptive when they should be critical and synthetic.

Authors' response:

We again appreciate this comment. The sections of the review describes the numerous classes of compounds of natural origin, so keeping in view the extensive reports published on natural agents having anti HIV activity still boosts to the researchers and pharmaceutical industry professionals who have the interest to synthesized the new chemical entities having natural product’s skeleton/scaffolds.

One would expect an in-depth overview of SAR and mechanistic insights, but it never really comes out in the review. Instead, we get many pages that only list numerous compounds with no description of mechanistic work, controls and assays used to determine antiviral activities of the natural compounds.

Authors' response:

We appreciate this comment. In our review we tried to describe the potent natural molecules diagrammatically along with their IC50 values and also enlightens the mechanistic view about molecule that it acts through which mechanism such reverse transcriptase inhibitor, fusion inhibitors, protease inhibitors or integrase inhibitors. For instance in alkaloid section the compound matairesinoside isolated from Symplocos setchuensi, it inhibits the HIV replication via acting on the viral replicated enzymes. The text part of in whole manuscript describes about the mechanistic work. In an another example under section Natural plants as antioxidants leads to the DNA damage and also stimulate a NF-κB factor which helps in the transcription. Similarly in terpenoids section the compounds obtained from Garcinia speciosa, garcisaterpenes A, C and theprotostanes. They inhibit the activity of HIV reverse transcriptase and thus stop the HIV life cycle.

HIV pathogenesis is naively written with numerous scientific errors. It would be much more exciting if the authors could give us an overview of what is known, methods used to perform a literature search of the compounds described, any evidence of clinical trials on such compounds, translational challenges to be overcome in moving forward natural products with anti-HIV activity, instead of telling the reader on how existing antiretroviral therapy are bad when they have saved millions of lives.

Authors' response:

We appreciate this comment. The survey was done on the PubMed, ScienceDirect and Scopus trademark of Elsevier, Scifinder–Chemical Abstracts Service from American Chemical Society. We also agree with the opinion of honorable reviewer that no doubt, existing synthetic antiretroviral therapy saved the millions of lives. In view of increased resistance to drugs.

Till date there is no vaccine or cure for HIV infection, and the efficacy of antiretroviral therapy, which combines two or three antiviral agents, targeting different steps of the virus replication cycle, can be compromised by the selection of strains resistant to one or multiple drug classes and current treatment-associated toxicity. However, these drugs have only limited or transient clinical benefit due to their severe side effects and the emergence of viral variants resistant to HIV-1 inhibitors.

We also appreciate to the honorable reviewer to provide any evidences of clinical trials of such compounds, (+)-Calanolide A has already been the subject of a phase II clinical study in healthy, HIV negative  individuals. Studies have demonstrated that (+) calanolide A has a favourable safety profile in both animal and human subjects. A data of compounds under clinical trials have also been included in final updated version.

Moreover, it has been recommended by the World Health Organization (WHO) that ethnomedicines and various other natural constituents should be orderly tested in contrast to HIV while they may produce more affordable and durable therapeutic agents

Therefore, the discovery of new antiviral agents with innovative modes of action or targets. In this respect, the identification of one molecule able to inhibit more than one viral function would provide significant advantages. Natural products provide an immeasurable wealth of active molecules, and a great number of new drugs have been originated from these compounds.

Hence, the development of new active lead, selective and safe HIV-1 inhibitors still remains a high priority for medical research. In addition, natural products or derivatives from old drugs also provide an alternative to treatment.

An important aspect to consider is are there any reports that perform direct comparison between herbal medicines and existing anti-HIV medicines to demonstrate considerable benefits or improvements?

Authors' response:

In view of increased resistance to existing drugs. However, these existing drugs have only limited or transient clinical benefit due to their severe side effects and the emergence of viral variants resistant to HIV-1 inhibitors. Whereas, Natural products provide an immeasurable wealth of active molecules, and a great number of new drugs have been originated from these compounds. Therefore, the development of new active leads, selective and safer therapeutic candidates still remains a high priority for medical research and scientific community.

Most sections of the review too are difficult to follow due to numerous grammatical errors. Overall, this review needs to be rewritten and focused.

Authors' response:

We appreciate this comment. As suggested by the honorable reviewer the final version of manuscript has been revised. The linguistic, grammatical and typographical errors have been removed, moreover the article have also been reviewed by native English expert.

Round 2

Reviewer 1 Report

I have found a few more minor mistakes:

Figure 4: the chirality of 17 is not correct in one of the centers

Figure 10: 53, the C should be capital

Figure 15: 88 and 89, the trans should be in italics.

I thank the authors for following my suggestions.

Reviewer 2 Report

Unfortunately my critiques were not addressed in the revision